# Towards a Unified Theoretical Understanding of Non-contrastive Learning via Rank Differential Mechanism

**Zhijian Zhuo**[1*]   **Yifei Wang**[1*]   **Jinwen Ma**[1]    **Yisen Wang**[2,3†]

[1]School of Mathematical Sciences, Peking University
[2]National Key Lab of General Artificial Intelligence,
   School of Intelligence Science and Technology, Peking University
[3]Institute for Artificial Intelligence, Peking University

## Abstract

Recently, a variety of methods under the name of non-contrastive learning (like BYOL, SimSiam, SwAV, DINO) show that when equipped with some asymmetric architectural designs, aligning positive pairs alone is sufficient to attain good performance in self-supervised visual learning. Despite some understandings of some specific modules (like the predictor in BYOL), there is yet no unified theoretical understanding of how these seemingly different asymmetric designs can all avoid feature collapse, particularly considering methods that also work without the predictor (like DINO). In this work, we propose a unified theoretical understanding for existing variants of non-contrastive learning. Our theory named Rank Differential Mechanism (RDM) shows that all these asymmetric designs create a consistent rank difference in their dual-branch output features. This rank difference will provably lead to an improvement of effective dimensionality and alleviate either complete or dimensional feature collapse. Different from previous theories, our RDM theory is applicable to different asymmetric designs (with and without the predictor), and thus can serve as a unified understanding of existing non-contrastive learning methods. Besides, our RDM theory also provides practical guidelines for designing many new non-contrastive variants. We show that these variants indeed achieve comparable performance to existing methods on benchmark datasets, and some of them even outperform the baselines. Our code is available at `https://github.com/PKU-ML/Rank-Differential-Mechanism`.

## 1 Introduction

Self-supervised learning of visual representations has undergone rapid progress in recent years, particularly due to the rise of contrastive learning (CL) (Oord et al., 2018; Wang et al., 2021). Canonical contrastive learning methods like SimCLR (Chen et al., 2020) and MoCo (He et al., 2020) utilize both positive samples (for feature alignment) and negative samples (for feature uniformity). Surprisingly, researchers notice that CL can also work well by only aligning positive samples, which is referred to as non-contrastive learning. Without the help of negative samples, various techniques are proposed to prevent feature collapse, for example, stop-gradient, momentum encoder, predictor (BYOL (Grill et al., 2020), SimSiam (Chen & He, 2021)), Sinkhorn iterations (SwAV (Caron et al., 2020)), feature centering and sharpening (DINO (Caron et al., 2021)). These above designs all create a certain of *asymmetry* between the online branch (with gradient) and the target branch (without gradient) (Wang et al., 2022a). Empirically, these tricks can successfully alleviate feature collapse and obtain comparable or even superior performance than canonical contrastive learning. Despite this progress, it is still not clear why these different heuristics can reach the same goal.

Some existing works are proposed to understand some specific non-contrastive techniques, mostly focusing on the predictor head proposed by BYOL (Grill et al., 2020). From an empirical side, Chen & He (2021) think that the predictor helps approximate the expectation over augmentations, and Zhang et al. (2022a) take a center-residual decomposition of representations for analyzing the collapse. From a theoretical perspective, Tian et al. (2021) analyze the dynamics of predictor weights

---

*Equal Contribution.
†Corresponding Author: Yisen Wang (yisen.wang@pku.edu.cn).

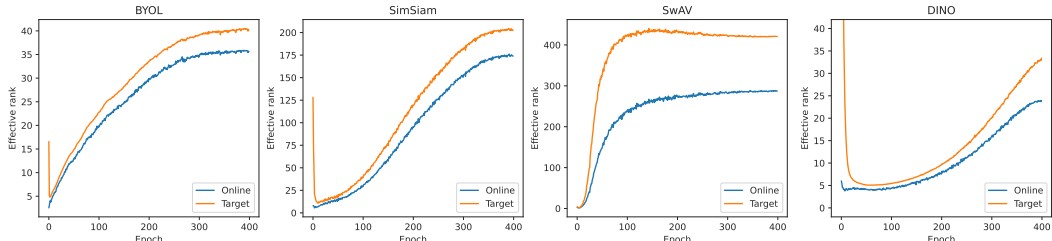

Figure 1: The effective rank of the normalized outputs of the online and target branches for four different non-contrastive methods (BYOL, SimSiam, SwAV, and DINO) on CIFAR-10.

under simple linear networks, and Wen & Li (2022) obtain optimization guarantees for two-layer nonlinear networks. These theoretical discussions often need strong assumptions on the data distribution (*e.g.,* standard normal (Tian et al., 2021)) and augmentations (*e.g.,* random masking (Wen & Li, 2022)). Besides, their analyses are often problem-specific, which is hardly extendable to other non-contrastive variants without a predictor, *e.g.,* DINO. Therefore, a natural question is raised here:

*Are there any basic principles behind these seemingly different techniques?*

In this paper, we make the first attempt in this direction by discovering a common mechanism behind these non-contrastive variants. To get a glimpse of it, in Figure 1, we measure the effective rank (Roy & Vetterli, 2007) of four different non-contrastive methods (BYOL, SimSiam, SwAV, and DINO). We find the following phenomenons: 1) among different methods, the target branch (orange line) has consistently higher rank than the online branch (blue line); 2) after the initial warmup stage, the rank of the online branch (blue line) consistently improves along the training process. Inspired by this observation, we propose a new theoretical understanding of non-contrastive methods, dubbed Rank Differential Mechanism (RDM), where we show that these different techniques essentially behave as a low-pass spectral filter, which is guaranteed to induce the rank difference above and avoid feature collapse along the training. We summarize the contribution of this work as follows:

- **Asymmetry matters for feature diversity**. In contrast to common beliefs, we show that even a symmetric architecture can provably alleviate complete feature collapse. However, it still suffers from low feature diversity, collapsing to a very low dimensional subspace. It indicates the key role of asymmetry is to avoid *dimensional* feature collapse.

- **Asymmetry induces low-pass filters that provably avoid dimensional collapse.** Based on theoretical and empirical evidence on real-world data, we point out the common underlying mechanism of asymmetric designs in BYOL, SimSiam, SwAV, DINO is that they behave as low-pass online-branch filters, or equivalently, high-pass target-branch filters. We further show that the asymmetry-induced low-pass filter can *provably* yield the rank collapse (Figure 1) and prevent feature collapse along the training process.

- **Principled designs of asymmetry.** Following the principle of RDM, we design a series of non-contrastive variants to empirically verify the effectiveness of our theory. For the online encoder, we show that different variants of low-pass filters can also attain fairly good performance. We also design a new kind of *target predictors with high-pass filters*. Experiments show that SimSiam with our target predictors can outperform DirectPred (Tian et al., 2021) and achieve comparable or even superior performance to the original SimSiam.

## 2 RELATED WORK

**Non-contrastive Learning**. Among existing methods, BYOL (Grill et al., 2020) is the first to show we can alleviate the feature collapse of aligning positive samples along with an online predictor and a momentum encoder. Later, SimSiam (Chen & He, 2021) further simplifies this requirement and shows that only the online predictor is enough. As for another thread, SwAV (Caron et al., 2020) applies Sinkhorn-Knopp iterations (Cuturi, 2013) to the target output from an optimal transport view. DINO (Caron et al., 2021) further simplifies this approach by simply combining feature centering and feature sharpening. Remarkably, all these methods adopt an online-target dual-branch architecture and gradients from the target branch are detached. Our theory provides a unified understanding of these designs and reveals their common underlying mechanisms. Additional comparison with related work is included in Appendix F.

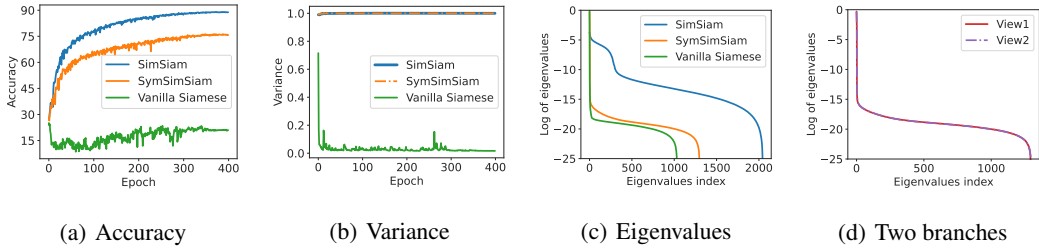

(a) Accuracy      (b) Variance      (c) Eigenvalues      (d) Two branches

Figure 2: Comparison of the asymmetric SimSiam with two symmetric baselines, SymSimSiam (ours) and Vanilla Siamese on CIFAR-10, where SymSimSiam adopts $\mathcal{L}_{\mathrm{sym}}(f'_\theta)$ (Eq. 2), while Vanilla Siamese adopts $\mathcal{L}_{\mathrm{sym}}(f_\theta)$. (a): Linear probing test accuracy. (b): Feature variance of normalized outputs. (c): Eigenvalues of the correlation matrix of the normalized outputs. (d): Eigenvalues of the correlation matrix of the normalized outputs from two branches of SymSimSiam.

**Dimensional Collapse of Self-supervised Representations.** Prior to ours, several works also explore the dimensional collapse issue in contrastive learning. Ermolov et al. (2021), Hua et al. (2021), Weng et al. (2022) and Zhang et al. (2022b) propose whitening techniques to alleviate dimensional collapse, similar in spirit to Barlow Twins (Zbontar et al., 2021) with a feature decorrelation regularization. Jing et al. (2022) point out the dimensional collapse of contrastive learning without using the projector, and propose DirectCLR as a direct replacement. Instead, our work mainly focuses on understanding the role of asymmetric designs on overcoming dimensional collapse.

**Theoretical Analysis on Contrastive Learning.** Saunshi et al. (2019) first establish downstream guarantees for contrastive learning, which are later gradually refined (Nozawa & Sato, 2021; Ash et al., 2022; Bao et al., 2022). Tosh et al. (2021) and Lee et al. (2021) also propose similar guarantees on downstream tasks. However, these methods mostly rely on the conditional independence assumption that is far from practice. Recently, HaoChen et al. (2021) and Wang et al. (2022b) propose an augmentation graph perspective with more practical assumptions, and contribute the generalization ability to the existence of augmentation overlap (which also exists for non-contrastive learning). Wen & Li (2021) analyze the feature dynamics of contrastive learning with shadow ReLU networks.

## 3   ASYMMETRY IS THE KEY TO ALLEVIATE DIMENSIONAL COLLAPSE

Prior works tend to believe that asymmetric designs are necessary for avoiding complete feature collapse (Zhang et al., 2022a), while we show that a *fully symmetric* architecture, dubbed SymSimSiam (Symmetric Simple Siamese network), can also avoid complete collapse. Specifically, we simply align the positive pair $(x, x^+)$ with a symmetric alignment loss,

$$\mathcal{L}_{\mathrm{sym}}(f'_\theta) = -\mathbb{E}_{x,x^+} f'_\theta(x)^\top f'_\theta(x^+), \tag{1}$$

where we apply feature centering on the output of an encoder $f$, *i.e.,* $f'_\theta(\cdot) = f_\theta(\cdot) - \mu$. The feature average $\mu = \mathbb{E}_x f_\theta(x)$ can be computed via mini-batch estimate or exponential moving average as in Batch Normalization (Ioffe & Szegedy, 2015) (see Algorithm 1). Theorem 1 states that SymSimSiam can avoid complete collapse by simultaneously maximizing feature variance $\mathrm{Var}(f_\theta(x))$.

**Theorem 1.** *When $f_\theta(x)$ is $\ell_2$-normalized, the SymSimSiam objective is equivalent to*

$$\mathcal{L}_{\mathrm{sym}}(f'_\theta) = \mathcal{L}_{\mathrm{sym}}(f_\theta) - \mathrm{Var}(f_\theta(x)) + 1 = -\mathbb{E}_{x,x^+} f_\theta(x)^\top f_\theta(x^+) - \mathbb{E}_x \|f_\theta(x) - \mu\|^2 + 1. \tag{2}$$

Empirically shown in Figures 2(a) & 2(b), compared to vanilla Siamese network, SymSimSiam indeed alleviates complete collapse, *i.e.,* the feature variance is maximized along training and a good linear probing accuracy is achieved. However, the accuracy of SymSimSiam is also obviously lower than the asymmetric SimSiam (Figure 2(a)). This indicates that symmetric design could alleviate complete collapse, but it may be *not* enough to prevent *dimensional feature collapse*. Intuitively, features uniformly distributed on a great circle of a unit sphere have maximal variance while being dimensionally collapsed. As further shown in Figure 2(c), SymSimSiam indeed suffers from more severe dimensional collapse compared to SimSiam. With few effective dimensions, the encoder network has limited ability to encode rich semantics (HaoChen et al., 2021).

The above SymSimSiam experiments show that with a symmetric architecture, we can easily prevent complete collapse while hardly improving the effective feature dimensionality for overcoming

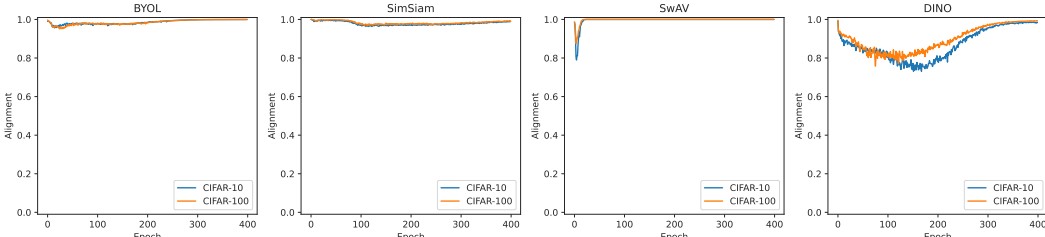

Figure 3: Eigenspace alignment of the online and target outputs along the training. Given $u_i$ as an eigenvector of $\mathbb{C}_z$, if it is also an eigenvector of $\mathbb{C}_p$, then $\mathbb{C}_p u_i = \lambda'_i u_i$ is in the same direction as $u_i$. Thus, we measure the alignment of $u_i$ by computing the cosine similarity between $u_i$ and $\mathbb{C}_p u_i$, and take the average over each $u_i$ as the overall eigenspace alignment (details in Appendix B.3).

dimensional collapse. Instead, we also notice that SimSiam with asymmetric designs can alleviate dimensional collapse and achieve better performance. It reflects the fact that the asymmetry in existing non-contrastive methods is the key to alleviating *dimensional collapse*, which leads us to the main focus of our paper, *i.e.,* demystifying asymmetric designs.

## 4 THE RANK DIFFERENCE MECHANISM OF ASYMMETRIC DESIGNS

In Figure 1, we have observed a common mechanism behind the non-contrastive learning methods: these asymmetric designs create *positive rank differences* between the target and online outputs consistently throughout training. Here, we provide a formal analysis of this phenomenon from both theoretical and empirical sides and show how it helps alleviate dimensional feature collapse.

**Problem Setup.** Given a set of natural training data $\bar{\mathcal{X}} = \{\bar{x} \mid \bar{x} \in \mathbb{R}^d\}$, we can draw a pair of positive samples $(x, x^+)$ from independent random augmentations of a natural example $\bar{x}$ with distribution $\mathcal{A}(\cdot|\bar{x})$. Their joint distribution satisfies $\mathcal{P}(x, x^+) = \mathcal{P}(x^+, x) = \mathbb{E}_{\bar{x}} \mathcal{A}(x|\bar{x}) \mathcal{A}(x^+|\bar{x})$. Without loss of generality, we consider a finite sample space $|\mathcal{X}| = n$ (can be exponentially large) following HaoChen et al. (2021), and denote the collection of online and target outputs as $p, z \in \mathbb{R}^{n \times k}$ whose $x$-th row is $p_x, z_x$, respectively. We consider a general alignment loss

$$L(p) = \mathbb{E}_{x,x^+} \ell(p_x, \mathrm{sg}(z_{x^+})) \tag{3}$$

to cover different variants of non-contrastive methods. The online and target outputs $p_x, z_x$ are either $\ell_2$-normalized (BYOL and SimSiam) or softmax-normalized (SwAV and DINO). For the loss function $\ell$, BYOL and SimSiam adopt the mean square error (MSE) loss, while SwAV and DINO adopt the cross entropy (CE) loss. $\mathrm{sg}(\cdot)$ denotes stopping the gradients from the operand. For an encoder $f$, we define its feature correlation matrix $\mathbb{C} = \mathbb{E}_x f(x) f(x)^\top$, whose spectral decomposition is $\mathbb{C} = V \Lambda V^\top$, where $V$ contains unit eigenvectors in columns, and $\Lambda$ is the diagonal matrix with descending eigenvalues $\lambda_1 \geq \cdots \geq \lambda_k \geq 0$.

**Measure of Dimensional Collapse.** A well-known measure of the effective feature dimensionality is the effective rank (erank) of the feature correlation matrix $\mathbb{C} = \mathbb{E}_x f(x) f(x)^\top$ (Roy & Vetterli, 2007). Specifically, $\mathrm{erank}(\mathbb{C}) = \exp(H(q))$, where $q = (q_1, \ldots, q_k), q_i = \lambda_i / \sum_i \lambda_i$ are the normalized eigenvalues as a probability distribution, and $H(q) = -\sum_i q_i \log(q_i)$ is its Shannon entropy. Compared to the canonical rank, the effective rank is real-valued and invariant to feature scaling. A more uniform distribution of eigenvalues has a larger effective rank, and vice versa. Thus, the effective rank of $\mathbb{C}$ is a proper metric to measure the degree of dimensional feature collapse.

**Spectral Filters.** In the signal processing literature, a spectral filtering process $G$ of a signal $f$ is to apply a scalar function (*i.e.,* a spectral filter) $g : \mathbb{R} \to \mathbb{R}$ element-wisely on its eigenvalues in its spectral domain, *i.e.,* $u_x = G f(x) = V g(\Lambda) V^\top f(x)$, where $G = V g(\Lambda) V^\top$ is also known as a spectral convolution operator. The filtered signal admits $C_u = \mathbb{E}_x u_x u_x^\top = V g(\Lambda) \Lambda V^T$. Depending on the property of $g$, a filter can be categorized as low-pass, high-pass, band-pass or band-stop. Generally speaking, a low-pass filter will amplify large eigenvalues and diminish smaller ones (*e.g.,* a monotonically increasing function), and a high-pass filter does the opposite. Many canonical algorithms can be seen as special cases of spectral filtering, *e.g.,* PCA-based image denoising is low-pass filtering.

## 4.1 Asymmetric Designs Behave as Spectral Filters

First of all, we notice that regardless of the existence of asymmetry, the alignment of two-branch outputs in non-contrastive learning will enforce the two-branch output features of the positive pairs to be close to each other. Therefore, from a spectral viewpoint, a natural hypothesis is that the online and target features will be aligned into the same eigenspace, and only differ slightly in the eigenvalues. We describe this hypothesis formally below.

**Definition 1** (Eigenspace alignment). *For two matrices $A$ and $B$, they have aligned eigenspace if*

$$\exists\, V, \; s.t. \; A = V\Lambda_a V^\top, B = V\Lambda_b V^\top, \tag{4}$$

*where $V$ is an orthogonal matrix of eigenvectors and $\Lambda_a, \Lambda_b$ consist non-increasing eigenvalues.*

**Hypothesis 1.** *During training, non-contrastive learning aligns the eigenspace of three correlation matrices of output features: the online correlation $\mathbb{C}_p = \mathbb{E}_x p_x p_x^\top$, the target correlation $\mathbb{C}_z = \mathbb{E}_x z_x z_x^\top$, and the feature correlation of positive samples $\mathbb{C}_+ = \mathbb{E}_{x,x^+} z_x z_{x^+}^\top$.*

Next, we validate this hypothesis from both theoretical and empirical aspects. To begin with, we consider a simplified setting adopted in prior work (Tian et al., 2021) for the ease of analysis: 1) data isotropy, where the natural data distribution $p(\bar{x})$ has zero mean and identity covariance, and the augmentation $\mathcal{A}(x|\bar{x})$ has mean $\bar{x}$ and covariance $\sigma^2 I$; 2) linear encoder $z_x = f(x) = W_f x, W_f \in \mathbb{R}^{d\times k}$; 3) linear online predictor $p_x = Wz_x, W \in \mathbb{R}^{k\times k}$. Under this setting, the following lemma shows that for an arbitrary encoder $f$, the eigenspace of three correlation matrices indeed align well:

**Lemma 1.** *With the assumptions above as in Tian et al. (2021), when the predictor $W^*$ minimizes the alignment loss (Eq. 3), we have*

$$\exists\, V, \; s.t. \; \mathbb{C}_p = V\Lambda_p V^\top, \mathbb{C}_z = V\Lambda_z V^\top, \mathbb{C}_+ = V\Lambda_+ V^\top, \tag{5}$$

*where $V$ is an orthogonal matrix and $\Lambda_p, \Lambda_z, \Lambda_+$ are diagonal matrices consisting of descending eigenvalues $\lambda_i^p, \lambda_i^z, \lambda_i^+, i = 1, \ldots, k$, respectively.*

Next, we provide an empirical examination of Hypothesis 1 on real-world data. From Figures 3, we can see that there is a consistently high degree of eigenspace alignment between $\mathbb{C}_p$ and $\mathbb{C}_z$ among all non-contrastive methods.[1] In Appendix E.1, we further verify the alignment *w.r.t.* $\mathbb{C}_+$. Therefore, these methods indeed attain a fairly high degree of eigenspace alignment.

**A Spectral Filter View.** As a result of eigenspace alignment, the alignment loss essentially works on mitigating the difference in *eigenvalues*. Therefore, we can take a spectral perspective on non-contrastive methods, where the asymmetric designs are *equivalent* to a spectral filtering process applied to the target output $z_x$, or a target spectral filtering process applied to the target output $p_x$. As the two cases are equivalent, we mainly take the online filter as an example in the discussion below.

**Lemma 2.** *Denote an online filter function $g : \lambda^z \to \lambda^g$ that satisfies $\lambda_i^g = \sqrt{\lambda_i^p/\lambda_i^z}, i = 1, \ldots, k$. We can apply a spectral filtering on $z_x$ with $g$, and get $\tilde{p}_x = W_g z_x, W_g = Vg(\Lambda_z)V^\top$. Then, we have $\mathbb{C}_p = \mathbb{C}_{\tilde{p}} = \mathbb{E}_x \tilde{p}_x \tilde{p}_x^\top$. In other words, $p_x, \tilde{p}_x$ have the same feature correlation.*

This spectral filter view reveals the key difference between symmetric and asymmetric designs in non-contrastive learning. In the symmetric case, the two branches yield almost equal eigenvalues (Figure 2(d)). Thus, the alignment loss will quickly diminish and the representations collapse dimensionally (Figure 2(c)). Instead, in asymmetric designs, asymmetric components create a difference in eigenvalues such that the target output generally has a higher rank than the online output (Figure 4(a)). Therefore, the alignment loss will not easily diminish (not necessarily decrease; see Figure 9). Instead, the alignment improves feature diversity in an implicit way, as we will show later.

## 4.2 Low-pass Property of Asymmetry-induced Spectral Filters

The discussion above reveals that a specific asymmetric design behaves as a spectral filter when applied to non-contrastive learning. To gain some insights for a unified understanding, we further investigate whether there is a common pattern behind the filters of different asymmetric designs.

To achieve this, we calculate and plot the corresponding online filter $g(\lambda^z) = \sqrt{\lambda_i^p/\lambda_i^z}$ of each non-contrastive method. From Figure 4(b), we find that the spectral filters indeed look very similar,

---

[1] As DINO adopts dynamic feature sharpening coefficients, there is a larger change of its eigenspace alignment compared to others. Nevertheless, the alignment degree is always above 0.7, which is relatively high.

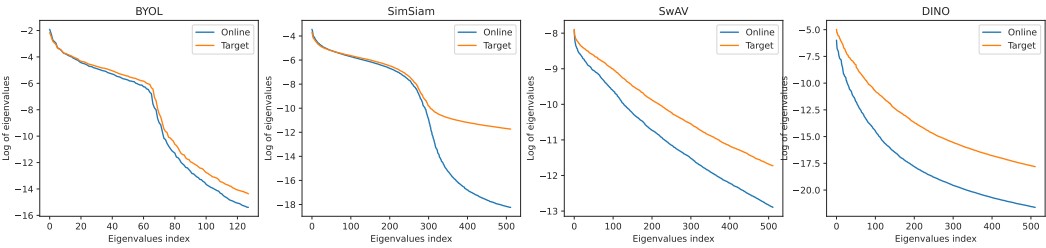

(a) Eigenvalues of the feature correction matrices of the online and target outputs.

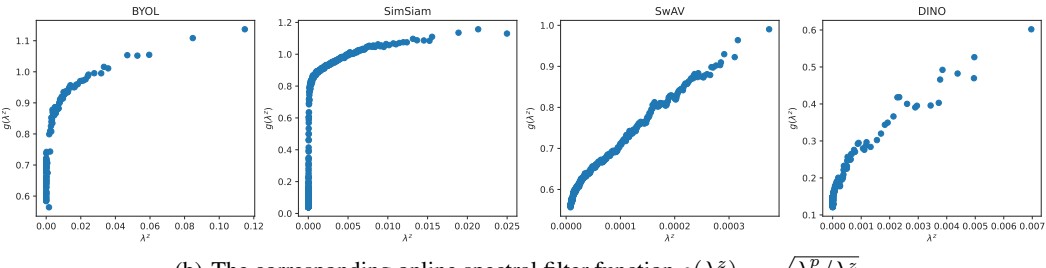

(b) The corresponding online spectral filter function $g(\lambda_i^z) = \sqrt{\lambda_i^p / \lambda_i^z}$.

Figure 4: Eigenvalues and spectral filters of each method on CIFAR-10: top eigenvalues (whose sum is larger than 99.99% of the total sum) are shown, 128 for BYOL and 512 for the other.

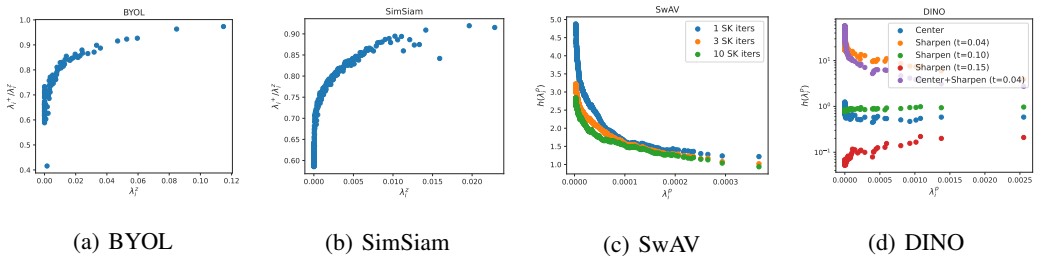

(a) BYOL      (b) SimSiam      (c) SwAV      (d) DINO

Figure 5: The role of asymmetric designs. (a) & (b): spectral filters of the ideal optimal predictors, $g^*(\lambda_i^z) = \lambda_i^+ / \lambda_i^z$ (Eq. 6). Both filters calculated from BYOL and SimSiam are almost monotonically decreasing. (c): The target filters using different Sinkhorn-Knopp (SK) iterations in SwAV. (d): The target filters using centering and/or sharpening with different target temperatures ($t$) in DINO (the online temperature is set to 0.1).

particularly in the sense that all filter functions are roughly monotonically increasing *w.r.t.* $\lambda^z$. This kind of filter is usually called a **low-pass filter** because it (relatively) enlarges low-frequency components (large eigenvalues) and shrinks high-frequency components (small eigenvalues). Based on this empirical finding, we propose the following hypothesis on the low-pass nature of asymmetry.

**Hypothesis 2.** *Asymmetric modules in non-contrastive learning behave as low-pass online filters. Formally, the corresponding spectral filter $g(\lambda^z) = \sqrt{\lambda_i^p / \lambda_i^z}$ is monotonically increasing.*[2]

We note that we are *not* suggesting *any* asymmetric design behaves as low-pass filters, as someone could easily apply a high-pass filter to the online output deliberately (which, as we have observed, will likely fail). Therefore, our hypothesis above only applies to asymmetric designs that work well in practice. In the discussion below, we further provide theoretical and empirical investigations of why existing asymmetric modules have a low-pass filtering effect.

**Case I. Online Predictor.** One popular kind of non-contrastive method, including BYOL and Sim-Siam, utilizes a learnable online predictor $g_\theta : \mathbb{R}^k \to \mathbb{R}^k$ for architectural asymmetry. One would wonder why such a *learnable* predictor will behave as a low-pass filter (Lemma 2). Here, we provide some theoretical insights in the following theorem.

---

[2]Equivalently, when viewing spectral filters from the target branch, the asymmetric modules behave as a high-pass target filter because the corresponding filter function $h(\lambda^p) = \sqrt{\lambda_i^z / \lambda_i^p}$ is monotonically decreasing.

**Theorem 2.** *Under Hypothesis 1, assume the invertibility of $\mathbb{C}_z$, the optimal predictor is given by*

$$W^* = \mathbb{C}_+ \mathbb{C}_z^{-1} = V\Omega V^\top, \quad \text{where } \omega_i = \Omega_{ii} = \lambda_i^+ / \lambda_i^z \in [0,1], i \in [k]. \tag{6}$$

*Therefore, the spectral property of the learnable filter is determined by the filter function $\lambda_i^+ / \lambda_i^z$.*

Theorem 2 shows that the predictor essentially learns to predict the correlation $\mathbb{C}_+$ between positive samples from one augmented view $\mathbb{C}_z$, *i.e.,* eliminating the augmentation noise and predicting the common features. The following lemma further reveals that 1) the correlation between positive samples is equivalent to the correlation of their underlying natural data $\bar{x}$, and 2) the difference between $\mathbb{C}_+$ and $\mathbb{C}_z$ is equal to the conditional covariance induced by data augmentation $\mathcal{A}(x|\bar{x})$.

**Lemma 3.** *The following equalities hold:*

1. *$\mathbb{C}_+ = \bar{\mathbb{C}} := \mathbb{E}_{\bar{x}} z_{\bar{x}} z_{\bar{x}}^\top$, where $z_{\bar{x}} = \mathbb{E}_{x|\bar{x}} z_x$;*

2. *$\mathbb{C}_z = \bar{\mathbb{C}} + \mathbb{V}_{x|\bar{x}}$, where $\mathbb{V}_{x|\bar{x}} = \mathbb{E}_{\bar{x}} \mathbb{E}_{x|\bar{x}} (z_x - z_{\bar{x}})(z_x - z_{\bar{x}})^\top$ is the conditional covariance.*

In practice, data augmentations mainly cause high-frequency noises in the feature space, therefore, the denoising predictor will behave as a low-pass filter. Indeed, Figures 5(a) and 5(b) empirically show that the filter derived from our theory, $\lambda_i^+ / \lambda_i^z$, aligns well with the actual learned predictor in Figure 4(b) as a low-pass filter.

**Case II. Target Transformation.** Another kind of non-contrastive method is to apply hand-crafted (instead of learned) transformations in the target branch, such as the Sinkhorn-Knopp (SK) iteration in SwAV (Caron et al., 2020) and centering-sharpening operators in DINO (Caron et al., 2021). In this case, we further study how these transformations behave as high-pass filters applied to the target output (see footnote of Hypothesis 2). Since it is generally hard to analyze these spatial transformations in the spectral domain, here we empirically study the role of each transformation on the resulting filter. Figure 5(c) shows that SK iterations indeed act like low-pass filters, and one iteration is enough. This explains why a few SK iterations already work well in SwAV. As for DINO, we notice that centering operation alone is not enough to produce a high-pass filter, which agrees with empirical results in DINO. Meanwhile, we notice that in order to obtain a high-pass filter (monotonically decreasing), it is necessary for DINO to apply a temperature smaller than the online branch ($< 0.1$), which is exactly the feature sharpening technique adopted in DINO (Caron et al., 2021). These facts show our theory aligns well with empirical results in non-contrastive learning.

### 4.3 Asymmetry-induced Low-pass Filters Save Non-contrastive Learning

In the discussion above, we observed a common pattern in existing asymmetric designs: their corresponding spectral filters are all low-pass. Here, we further show that this property is so essential that it can *provably* save non-contrastive learning from the risk of feature collapse by producing the rank difference between outputs and alleviating dimensional collapse during training.

First, let us take a look at its effect on the effective rank of two-branch output features. In the theorem below, we show that low-pass online filters are guaranteed to produce a consistently higher effective rank of target features than that of online features, as shown in Figure 1.

**Theorem 3.** *If $g(\lambda_i^z) = \sqrt{\lambda_i^p / \lambda_i^z}$ is monotonically increasing, or equivalently, $h(\lambda_i^p) = \sqrt{\lambda_i^z / \lambda_i^p}$ is monotonically decreasing, we have $\operatorname{erank}(\mathbb{C}_p) \leq \operatorname{erank}(\mathbb{C}_z)$. Further, as long as $g(\lambda_i^z)$ or $h(\lambda_i^p)$ is non-constant, the inequality holds strictly, $\operatorname{erank}(\mathbb{C}_p) < \operatorname{erank}(\mathbb{C}_z)$.*

Meanwhile, we also observe that for each output, its own effective rank is also successfully elevated along this process. This is not a coincidence. Below, we theoretically show how the rank difference alleviates dimensional collapse. As the training dynamics of deep neural networks is generally hard to analyze, prior works (Tian et al., 2021; Wen & Li, 2021) mainly deal with linear or shadow networks with strong assumptions on data distribution, which could be far from practice. As over-parameterized deep neural networks are very expressive, we instead adopt the unconstrained feature setting (Mixon et al., 2022; HaoChen et al., 2021) and consider gradient descent directly in the feature space $\mathbb{R}^k$. Taking the MSE loss $\ell(u,v) = \frac{1}{2}\|u-v\|^2$ as an example, the following theorem shows that the rank difference indeed helps improve the effective rank of online output $p$.

**Theorem 4.** *Under Hypothesis 1, when we apply an online spectral filter $p_x = W z_x$ (Lemma 2), gradient descent with step size $0 < \alpha < 1$ gives the following update at the $t$-th step,*

$$\lambda_{i,t+1}^p = \lambda_{i,t}^p \left( (1-\alpha)^2 + \alpha^2 h^2(\lambda_{i,t}^p) + 2\alpha(1-\alpha)h(\lambda_{i,t}^p)\frac{\lambda_{i,t}^+}{\lambda_{i,t}^z} \right), \ i = 1,\dots,k, \tag{7}$$

Table 1: Linear probing accuracy (%) of SimSiam with different **online** predictors (including learnable nonlinear and linear predictors of SimSiam) on CIFAR-10, CIFAR-100 and ImageNet-100.

| Predictor | CIFAR-10 | | | CIFAR-100 | | | IN-100 (400ep) | |
| --- | --- | --- | --- | --- | --- | --- | --- | --- |
| | 100ep | 200ep | 400ep | 100ep | 200ep | 400ep | acc@1 | acc@5 |
| SimSiam (nonlinear) | 74.00 | 84.39 | 89.41 | 46.12 | 55.51 | 63.39 | 77.96 | 94.26 |
| SimSiam (linear) | 72.42 | **83.67** | **88.9** | 42.98 | **55.69** | **62.02** | 77.18 | 94.02 |
| $g(\sigma) = \sigma$ | 73.91 | 82.74 | 86.51 | **46.73** | 54.45 | 59.59 | 75.06 | 93.28 |
| $g(\sigma) = \log(\sigma)$ | 74.80 | 80.72 | 87.52 | 39.82 | 48.59 | 58.08 | 77.22 | **94.34** |
| $g(\sigma) = \log(1 + \sigma)$ | **75.70** | 80.47 | 86.76 | 40.53 | 47.68 | 61.15 | **77.42** | 93.90 |
| $g(\sigma) = \log(1 + \sigma^2)$ | 75.51 | 82.28 | 87.51 | 39.19 | 48.97 | 59.06 | 77.22 | 94.00 |

Table 2: Linear probing accuracy (%) of SimSiam with different **target** predictors (including learnable nonlinear and linear predictors) on CIFAR-10, CIFAR-100 and ImageNet-100.

| Predictor | CIFAR-10 | | | CIFAR-100 | | | IN-100 (400ep) | |
| --- | --- | --- | --- | --- | --- | --- | --- | --- |
| | 100ep | 200ep | 400ep | 100ep | 200ep | 400ep | acc@1 | acc@5 |
| SimSiam (nonlinear) | 74.00 | 84.39 | 89.41 | 46.12 | 55.51 | 63.39 | 77.96 | 94.26 |
| SimSiam (linear) | 72.42 | 83.67 | **88.90** | 42.98 | 55.69 | 62.02 | 77.18 | 94.02 |
| $h(\sigma) = \sigma^{-0.3}$ | 77.33 | 85.57 | 88.74 | 48.07 | 58.07 | **62.83** | **77.68** | 94.16 |
| $h(\sigma) = \sigma^{-0.5}$ | **78.17** | 85.52 | 88.43 | 50.61 | 58.68 | 62.79 | 76.74 | 94.10 |
| $h(\sigma) = \sigma^{-0.7}$ | 77.98 | 85.79 | 88.45 | 51.04 | 58.37 | 62.69 | 77.28 | **94.94** |
| $h(\sigma) = \sigma^{-1}$ | 77.33 | **86.44** | 88.50 | **51.05** | **58.87** | 61.92 | 77.00 | 93.32 |

*where $h(\lambda) = 1/g(\lambda)$ is a high-pass filter because $g(\lambda)$ is a low-pass filter (Hypothesis 2). Then, the update $\lambda^p_{i,t+1}/\lambda^p_{i,t}$ will nicely correspond to a high-pass filter under either of the two conditions:*

1. *the learned encoder is nearly optimal, i.e., $W \approx W^*$ in Theorem 2.*

2. *$\lambda^+_{i,t} \approx \lambda^z_{i,t}$, which naturally holds with good positive alignment, i.e., $z_x \approx z_{x^+}$.*

*Then, according to Theorem 3, we have $\mathrm{erank}(\mathbb{C}_{p^{(t+1)}}) > \mathrm{erank}(\mathbb{C}_{p^{(t)}})$. In other words, the effective rank of online output will keep improving after gradient descent.*

Intuitively, the improvement of effective rank is a natural consequence of the rank difference. As the online output has a lower effective rank than the target output, optimizing its alignment loss *w.r.t.* the gradient-detached target $\mathrm{sg}(z_x)$ will enforce the online output $p_x$ to improve its effective rank in order to match the target output $z_x$. In this way, the rank difference becomes a ladder (created by asymmetric designs) for non-contrastive methods to gradually improve its effective feature dimensionality and get rid of dimensional feature collapse eventually.

**Note on Stop Gradient.** Our analysis above also reveals the importance of the stop gradient operation. In particular, when gradients from the target branch are not detached, in order to match the rank of two outputs, minimizing the alignment loss can also be fulfilled by simply *pulling down* the rank of the target output. In this case, the feature rank will never be improved without stop gradient.

## 5 PRINCIPLED ASYMMETRIC DESIGNS BASED ON RDM

The discussions above show that the rank differential mechanism provides a unified theoretical understanding of different non-contrastive methods. Besides, it also provides a general recipe for designing *new* non-contrastive variants. As Theorem 3 points out, the key requirement is that the asymmetry can produce a low-pass filter on the online branch, or equivalently, a high-pass filter on the target branch. Below, we propose some new variants directly following this principle.

### 5.1 VARIANTS OF ONLINE LOW-PASS FILTERS

Despite the learnable predictor (Section 4.2), we can also directly design online predictors with fixed low-pass filters. For numerical stability, we adopt the singular value decomposition (SVD) of the output to compute the eigenvalues and the eigenspace, *e.g.*, $z = U\Sigma^z V^\top$ with singular values $\sigma^z_i$'s. As $\lambda^z_i = (\sigma^{z_i})^2$, a filter that is monotone in $\lambda$ is also monotone in $\sigma$, and vice versa. Specifically, for

Table 3: Linear probing accuracy (%) of SimSiam with different predictors on ImageNet-1k.

| SimSiam (Learnable Online Filter) | | Online Low-pass Filter (ours) $g(\sigma) = \log(1 + \sigma)$ | | Target High-pass Filter (ours) $g(\sigma) = \sigma^{-0.3}$ | |
|---|---|---|---|---|---|
| acc@1 | acc@5 | acc@1 | acc@5 | acc@1 | acc@5 |
| 67.97 | 88.17 | 64.67 | 86.13 | 67.73 | 88.05 |

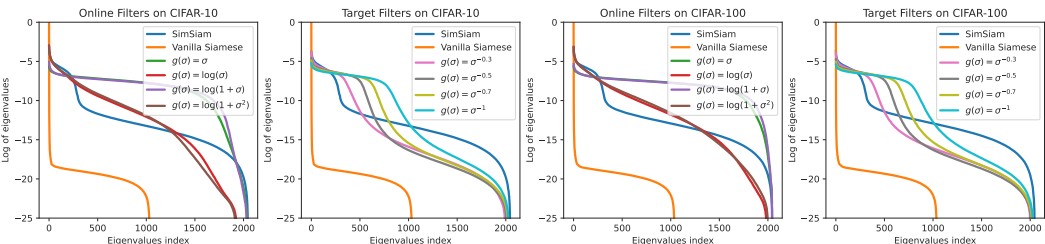

Figure 6: Eigenvalues of the correlation matrix of the normalized outputs for different variants of filters on CIFAR-10 and CIFAR-100.

an online encoder $f : \mathcal{X} \to \mathbb{R}^k$, we assign $W = V g(\Sigma^z) V^\top$, where $g(\Sigma^z)_{ii} = g(\sigma_i^z)$ is a low-pass filter that is monotonically increasing *w.r.t.* $\sigma$. We note that DirectPred proposed by Tian et al. (2021) is a special case with $g_0(\sigma) = \sigma$. Additionally, we consider three variants: 1) $g_1(\sigma) = \log(\sigma)$; 2) $g_2(\lambda) = \log(\sigma + 1)$; 3) $g_3(\sigma) = \log(\sigma^2 + 1)$. These three new variants are low-pass filters because they are monotonically increasing with $\sigma \geq 0$.

We evaluate their performance on four datasets, *i.e.,* CIFAR-10, CIFAR-100 (Krizhevsky et al., 2009), ImageNet-100 and ImageNet-1k (Deng et al., 2009). We use ResNet-18 (He et al., 2016) as the backbone encoder for CIFAR-10, CIFAR-100 and ImageNet-100 and adopt ResNet-50 for Imagenet-1k following standard practice. And the projector is a three-layer MLP in which BN (Ioffe & Szegedy, 2015) is applied to all layers. We adopt the linear probing task for evaluating the learned representations. More details are included in Appendix B.2. Table 1 shows that our designs of online predictors with different low-pass filters work well in practice, and achieve comparable performance to SimSiam with learnable predictors. In particular, it also significantly outperforms SymSimSiam (Figure 2(a)), showing that rank differences indeed help alleviate dimensional collapse (Figure 6).

## 5.2 VARIANTS OF TARGET HIGH-PASS FILTERS

In turn, we also consider applying a high-pass filter on the target branch using a *target predictor*. Compared to the online predictors above, target predictors have additional advantages: with stop gradient, we do not require backpropagation through SVD, which could reduce time overhead (Table 5). Specifically, we consider polynomial high-pass target filters $h(\sigma) = \sigma^p$ with different $-1 \leq p < 0$, which are all monotonically decreasing (see Algorithm 2).

We further evaluate the target filters following the same protocols above. As shown in Table 2, our high-pass target filters can often outperform SimSiam by a large margin with a relatively short training time. Especially, on CIFAR-10, the filter $h(\sigma) = \sigma^{-0.5}$ improves the baseline by 5.75% with 100 epochs and the filter $h(\sigma) = \sigma^{-1}$ improves the baseline by 2.77% with 200 epochs. Notably, on CIFAR-100, our methods outperform the baseline by a large margin (8.07%, 3.18%, and 0.81% improvements with 100, 200, and 400 epochs, respectively). Additional results based on the BYOL framework can be found in Appendix C.

## 6 CONCLUSION

In this paper, we presented a unified theoretical understanding of non-contrastive learning via the rank differential hypothesis. In particular, we showed that existing non-contrastive learning all produce a consistent rank difference between the online and the target outputs. Digging deeper into this phenomenon, we theoretically proved that low-pass online filters can yield such a rank difference and improve the effective feature dimensionality along the training. Meanwhile, we provided theoretical and empirical insights on how existing asymmetric designs produce low-pass filters. At last, following the principle of our theory, we designed a series of new online and target filters, and showed that they achieve comparable or even superior to existing asymmetric designs.

ACKNOWLEDGMENTS

Jinwen Ma is supported by the National Key Research and Development Program of China under grant 2018AAA0100205. Yisen Wang is partially supported by the National Natural Science Foundation of China (62006153), Open Research Projects of Zhejiang Lab (No. 2022RC0AB05), and Huawei Technologies Inc.

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

## A OMITTED PROOFS

In this section, we present proofs for all lemmas and theorems in the main paper.

### A.1 PROOF OF THEOREM 1

*Proof.* Since the output of $f_\theta$ is $\ell_2$-normalized, and $\mathbb{E}_x f(x) = \mathbb{E}_{x^+} f(x^+) = \mu$, we have $\mathcal{L}_{\text{sym}}(\theta) = -\mathbb{E}_{x,x^+} f'_\theta(x)^\top f'_\theta(x^+) = -\mathbb{E}_{x,x^+}(f_\theta(x) - \mu)^\top (f_\theta(x^+) - \mu) = -\mathbb{E}_{x,x^+} f_\theta(x)^\top f_\theta(x^+) + \|\mu\|^2$, and $\text{Var}(f(x)) = \mathbb{E}_x \|f(x)\|^2 - \|\mu\|^2 = 1 - \|\mu\|^2$. Thus, we conclude that $\mathcal{L}_{\text{sym}}(\theta) = -\mathbb{E}_{x,x^+} f_\theta(x)^\top f_\theta(x^+) - \text{Var}(f(x)) + 1$. $\square$

### A.2 PROOF OF LEMMA 1

*Proof.* Let $z_x = W_f x$ and $z_{x^+} = W_f x^+$, the loss functions is

$$\mathcal{L}(W_f, W) = \mathbb{E}_{x,x^+} \frac{1}{2} \|W z_x - \text{sg}(z_{x^+})\|^2 \tag{8}$$

$$= \mathbb{E}_{x,x^+} \frac{1}{2} \left(W z_x - \text{sg}(z_{x^+})\right)^\top \left(W z_x - \text{sg}(z_{x^+})\right) \tag{9}$$

$$= \mathbb{E}_{x,x^+} \frac{1}{2} \left(z_x^\top W^\top W z_x - \text{sg}(z_{x^+})^\top W z_x - z_x^\top W^\top \text{sg}(z_{x^+}) + \text{sg}(z_{x^+})^\top \text{sg}(z_{x^+})\right) \tag{10}$$

$$= \frac{1}{2} \mathbb{E}_{x,x^+} \left[\text{Tr}\left(z_x^\top W^\top W z_x\right) - 2 \text{Tr}\left(\text{sg}(z_{x^+})^\top W z_x\right) + \text{Tr}\left(\text{sg}(z_{x^+})^\top \text{sg}(z_{x^+})\right)\right] \tag{11}$$

$$= \frac{1}{2} \mathbb{E}_{x,x^+} \left[\text{Tr}\left(W^\top W z_x z_x^\top\right) - 2 \text{Tr}\left(W z_x \text{sg}(z_{x^+})^\top\right) + \text{Tr}\left(\text{sg}(z_{x^+}) \text{sg}(z_{x^+})^\top\right)\right] \tag{12}$$

$$= \frac{1}{2} \left[\text{Tr}\left(W^\top W \mathbb{E}_x z_x z_x^\top\right) - 2 \text{Tr}\left(W \mathbb{E}_{x,x^+} z_x z_{x^+}^\top\right) + \text{Tr}\left(\mathbb{E}_{x^+} z_{x^+} z_{x^+}^\top\right)\right]. \tag{13}$$

Notice that

$$\mathbb{C}_z = \mathbb{E}_x z_x z_x^\top = \mathbb{E}_{x^+} z_{x^+} z_{x^+}^\top \tag{14}$$

$$= \mathbb{E}_{\bar{x}} \mathbb{E}_{x|\bar{x}} W_f x \left(W_f x\right)^\top \tag{15}$$

$$= W_f \mathbb{E}_{\bar{x}} \mathbb{E}_{x|\bar{x}} x x^\top W_f^\top \tag{16}$$

$$= W_f \mathbb{E}_{\bar{x}} \left(\bar{x}\bar{x}^\top + \sigma^2 I\right) W_f^\top \tag{17}$$

$$= (1 + \sigma^2) W_f W_f^\top, \tag{18}$$

and

$$C_+ = \mathbb{E}_{x,x^+} z_x z_{x^+}^\top \tag{19}$$

$$= \mathbb{E}_{\bar{x}} \mathbb{E}_{x,x^+|\bar{x}} z_x z_{x^+}^\top \tag{20}$$

$$= \mathbb{E}_{\bar{x}} \left(\mathbb{E}_{x|\bar{x}} W_f x\right) \left(\mathbb{E}_{x|\bar{x}^+} W_f x^+\right)^\top \tag{21}$$

$$= W_f \mathbb{E}_{\bar{x}} \bar{x}\bar{x}^\top W_f^\top \tag{22}$$

$$= W_f W_f^\top. \tag{23}$$

Hence,

$$\mathcal{L}(W_f, W) = \frac{1}{2} \left[(1 + \sigma^2) \text{Tr}\left(W^\top W W_f W_f^\top\right) - 2 \text{Tr}\left(W W_f W_f^\top\right) + (1 + \sigma^2) \text{Tr}\left(W_f W_f^\top\right)\right]. \tag{24}$$

Taking partial derivative with respect to $W$, we get

$$\frac{\partial \mathcal{L}(W_f, W)}{\partial W} = (1 + \sigma^2) W W_f W_f^\top - W_f W_f^\top. \tag{25}$$

With additional weight decay, we have

$$\frac{\partial \mathcal{L}(W_f, W)}{\partial W} = (1 + \sigma^2) W W_f W_f^\top - W_f W_f^\top + \eta W, \tag{26}$$

where $\eta > 0$ is the coefficient of weight decay. Suppose the spectral decomposition $W_f W_f^\top$ is $W_f W_f^\top = V \Lambda V^\top$, where $V$ is an orthogonal matrix and $\Lambda$ is a diagonal matrix consisting of decending eigenvalues $\lambda_1, \cdots, \lambda_k$. Therefore,

$$\mathbb{C}_z = (1 + \sigma^2) W_f W_f^\top = V(1 + \sigma^2)\Lambda V^\top = V \Lambda_z V^\top, \tag{27}$$

$$\mathbb{C}_+ = \mathbb{E}_{x,x^+} z_x z_x^\top = W_f W_f^\top = V \Lambda V^\top. \tag{28}$$

Let $\frac{\partial \mathcal{L}(W)}{\partial W} = 0$, we get $W^* = W_f W_f^\top \left( (1 + \sigma^2) W_f W_f^\top + \eta I \right)^{-1} = V \Lambda_w V^\top$, where $\Lambda_w = \mathrm{diag}\{\lambda_1/((1 + \sigma^2)\lambda_1 + \eta), \cdots, \lambda_k/((1 + \sigma^2)\lambda_k + \eta)\}$. It follows that

$$\mathbb{C}_p = \mathbb{E}_x p_x p_x^\top = W^* \mathbb{E}_x z_x z_x^\top W^{*\top} = W^* \mathbb{C}_z W^{*\top} = V \Lambda_w^2 \Lambda_z V^\top = V \Lambda_p V^\top. \tag{29}$$

Hence, $\mathbb{C}_p, \mathbb{C}_z, \mathbb{C}_+$ share the same eigenspace. $\qquad\square$

### A.3 PROOF OF LEMMA 2

*Proof.* By the definition of $\tilde{p}_x = W_g z_x$, we know that

$$\mathbb{C}_{\tilde{p}} = \mathbb{E}_x \tilde{p}_x \tilde{p}_x^\top = \mathbb{E}_x W_g z_x z_x^\top W_g^\top = W_g \left( \mathbb{E}_x z_x z_x^\top \right) W_g^\top = W_g \mathbb{C}_z W_g^\top. \tag{30}$$

Since $W_g = V g(\Lambda_z) V^\top, \mathbb{C}_z = V \Lambda_z V^\top, \mathbb{C}_p = V \Lambda_p V^\top$ and $g(\lambda_i^z) = \sqrt{\lambda_i^p / \lambda_i^z}$, we have

$$\mathbb{C}_{\tilde{p}} = V g(\Lambda_z) V^\top V \Lambda_z V^\top V g(\Lambda_z) V^\top = V g(\Lambda_z) \Lambda_z g(\Lambda_z) V^\top = V \Lambda_p V^\top = \mathbb{C}_p. \tag{31}$$

$$\square$$

### A.4 PROOF OF LEMMA 3

*Proof.* For the first equality, we have

$$\mathbb{C}_+ = \mathbb{E}_{x,x^+} z_x z_{x^+}^\top = \mathbb{E}_{\bar{x}} \mathbb{E}_{x,x^+|\bar{x}} z_x z_{x^+}^\top = \mathbb{E}_{\bar{x}} \left( \mathbb{E}_{x|\bar{x}} z_x \right) \left( \mathbb{E}_{x|\bar{x}^+} z_{x^+} \right)^\top = \mathbb{E}_{\bar{x}} z_{\bar{x}} z_{\bar{x}}^\top = \bar{\mathbb{C}}. \tag{32}$$

For the first part,

$$\bar{\mathbb{C}} + \mathbb{V}_{x|\bar{x}} = \mathbb{E}_{\bar{x}} z_{\bar{x}} z_{\bar{x}}^\top + \mathbb{E}_{\bar{x}} \left[ \mathbb{E}_{x|\bar{x}} \left( z_x - z_{\bar{x}} \right) \left( z_x - z_{\bar{x}} \right)^\top \right] \tag{33}$$

$$= \mathbb{E}_{\bar{x}} z_{\bar{x}} z_{\bar{x}}^\top + \mathbb{E}_{\bar{x}} \left[ \mathbb{E}_{x|\bar{x}} z_x z_x^\top - \left( \mathbb{E}_{x|\bar{x}} z_x \right) z_{\bar{x}}^\top - z_{\bar{x}} \left( \mathbb{E}_{x|\bar{x}} z_x \right)^\top + z_{\bar{x}} z_{\bar{x}}^\top \right] \tag{34}$$

$$= \mathbb{E}_{\bar{x}} z_{\bar{x}} z_{\bar{x}}^\top + \mathbb{E}_{\bar{x}} \mathbb{E}_{x|\bar{x}} z_x z_x^\top - \mathbb{E}_{\bar{x}} z_{\bar{x}} z_{\bar{x}}^\top \tag{35}$$

$$= \mathbb{E}_x z_x z_x^\top \tag{36}$$

$$= \mathbb{C}_z. \tag{37}$$

$$\square$$

### A.5 PROOF OF THEOREM 2

*Proof.* Similar to the derivation in the proof of Lemma 1, we have

$$\mathcal{L}(W) = \mathbb{E}_{x,x^+} \frac{1}{2} \| W z_x - \mathrm{sg}(z_{x^+}) \|^2 \tag{38}$$

$$= \frac{1}{2} \left[ \mathrm{Tr} \left( W^\top W \mathbb{E}_x z_x z_x^\top \right) - 2 \mathrm{Tr} \left( W \mathbb{E}_{x,x^+} z_x z_{x^+}^\top \right) + \mathrm{Tr} \left( \mathbb{E}_{x^+} z_{x^+} z_{x^+}^\top \right) \right]. \tag{39}$$

Notice that $\mathbb{C}_z = \mathbb{E}_x z_x z_x^\top = \mathbb{E}_{x^+} z_{x^+} z_{x^+}^\top$ and

$$\mathbb{E}_{x,x^+} z_x z_{x^+}^\top = \mathbb{E}_{\bar{x}} \mathbb{E}_{x,x^+|\bar{x}} z_x z_{x^+}^\top \tag{40}$$

$$= \mathbb{E}_{\bar{x}} \left( \mathbb{E}_{x|\bar{x}} z_x \right) \left( \mathbb{E}_{x|\bar{x}^+} z_{x^+} \right)^\top \tag{41}$$

$$= \mathbb{E}_{\bar{x}} z_{\bar{x}} z_{\bar{x}}^\top \tag{42}$$

$$= \mathbb{C}_{\bar{z}}. \tag{43}$$

Hence,

$$\mathcal{L}(W) = \frac{1}{2} \left[ \text{Tr} \left( W^\top W \mathbb{C}_z \right) - 2 \text{Tr} \left( W \bar{\mathbb{C}} \right) + \text{Tr} \left( \mathbb{C}_z \right) \right]. \tag{44}$$

Taking partial derivative with respect to $W$, we get

$$\frac{\partial \mathcal{L}(W)}{\partial W} = W \mathbb{C}_z - \bar{\mathbb{C}}. \tag{45}$$

Let $\frac{\partial \mathcal{L}(W)}{\partial W} = 0$, we have $W^* = \bar{\mathbb{C}} \mathbb{C}_z^{-1}$. Since $\mathbb{C}_z, \bar{\mathbb{C}}$ have aligned eigenspace $V$, we have

$$V \bar{\Lambda} V^\top + \mathbb{V}_{x|\bar{x}} = V \Lambda_z V^\top \tag{46}$$

$$\Longrightarrow \mathbb{V}_{x|\bar{x}} = V \left( \Lambda_z - \bar{\Lambda} \right) V^\top, \tag{47}$$

which implies that the corresponding eigenvalues satisfy $\varepsilon_i = \lambda_i^z - \bar{\lambda}_i \geq 0$, where $\bar{\lambda}_i, \varepsilon_i$ denote the $i$-th eigenvalues of $\bar{\mathbb{C}}, \mathbb{V}_{x|\bar{x}}$, respectively. And $\bar{\mathbb{C}} \mathbb{C}_z^{-1} = V \bar{\Lambda} V^\top \left( V \Lambda_z V^\top \right)^{-1} = V \bar{\Lambda} \Lambda_z^{-1} V^\top = V \Omega V^\top$, where $\Omega$ is a diagonal matrix and $\Omega_{ii} = \frac{\bar{\lambda}_i}{\lambda_i^z} \in [0,1], i = 1, 2, \ldots, k$. $\qquad\square$

### A.6 PROOF OF THEOREM 3

Before the proof of Theorem 3, we first introduce the following useful lemma.

**Lemma 4.** *Assume that $\sum_{i=1}^k q_i = 1$ and $q_1 \geq q_2 \geq \cdots \geq q_k > 0$, then for any $1 \leq i < j \leq k$ and any $\Delta \in (0, p_j)$, it holds that*

$$H(q_1, q_2, \ldots, q_k) > H(q_1, \ldots, q_i + \Delta, \ldots, q_j - \Delta, \ldots, q_k). \tag{48}$$

*Proof.* We first note that

$$H(q_1, q_2, \ldots, q_k) > H(q_1, \ldots, q_i + \Delta, \ldots, q_j - \Delta, \ldots, q_k) \tag{49}$$

$$\Longleftrightarrow -q_i \log(q_i) - q_j \log(q_j) > -(q_i + \Delta) \log(q_i + \Delta) - (q_j - \Delta) \log(q_j - \Delta). \tag{50}$$

Define $f(x) = -(q_i + x) \log(q_i + x) - (q_j - x) \log(q_j - x)$ and its first order derivative satisfies

$$f'(x) = -\log(q_i + x) + \log(q_j - x) < 0, \quad \forall x \in (0, q_j). \tag{51}$$

Hence, Equation (50) holds. This completes the proof of the lemma. $\qquad\square$

*Proof of Theorem 3.* Let $q_i^z = \lambda_i^z / (\sum_{l=1}^k \lambda_l^z)$ and $q_i^p = \lambda_i^p / (\sum_{l=1}^k \lambda_l^p)$, where $i = 1, 2, \ldots, k$, then $\sum_{i=1}^k q_i^z = \sum_{i=1}^k q_i^p = 1$, $q_1^z \geq q_2^z \geq \cdots \geq q_k^z$ and $q_1^p \geq q_2^p \geq \cdots \geq q_k^z$. Without loss of generality, we assume that $q_k^p, q_k^z > 0$. Because $g(\lambda_i^z) = \sqrt{\lambda_i^p / \lambda_i^z}$ is monotonically increasing and $\lambda_i^z \geq \lambda_j^z$, for any $1 \leq i < j \leq k$, we have

$$\frac{q_i^p}{q_j^p} = \frac{\lambda_i^p / (\sum_{l=1}^k \lambda_l^p)}{\lambda_j^p / (\sum_{l=1}^k \lambda_l^p)} = \frac{\lambda_i^p}{\lambda_j^p} = \frac{g^2(\lambda_i^z) \lambda_i^z}{g^2(\lambda_j^z) \lambda_j^z} \geq \frac{\lambda_i^z}{\lambda_j^z} = \frac{\lambda_i^z / (\sum_{l=1}^k \lambda_l^z)}{\lambda_j^z / (\sum_{l=1}^k \lambda_l^z)} = \frac{q_i^z}{q_j^z}. \tag{52}$$

If $g(\lambda_i^z)$ is constant, it follows that

$$\frac{q_i^p}{q_j^p} = \frac{q_i^z}{q_j^z}, \quad \forall 1 \leq i < j \leq k. \tag{53}$$

Combining with $\sum_{i=1}^k q_i^z = \sum_{i=1}^k q_i^p = 1$, we get $q_i^p = q_i^z, i = 1, 2, \ldots, k$. Hence,

$$H(q_1^p, q_2^p, \ldots, q_k^p) = H(q_1^z, q_2^z, \ldots, q_k^z), \tag{54}$$

which implies that $\text{erank}(\mathbb{C}_p) = \text{erank}(\mathbb{C}_z)$.

If $g(\lambda_i^z)$ is non-constant, then $g(\lambda_1^z) > g(\lambda_k^z)$. And it follows that

$$\frac{q_1^p}{q_k^p} > \frac{q_1^z}{q_k^z}. \tag{55}$$

Arming with Equations (52) and (55), we get

$$1 = \sum_{i=1}^{k} q_i^p = q_k^p \sum_{i=1}^{k} \frac{q_i^p}{q_k^p} > q_k^p \sum_{i=1}^{k} \frac{q_i^z}{q_k^z} = \frac{q_k^p}{q_k^z} \sum_{i=1}^{k} q_i^z = \frac{q_k^p}{q_k^z}, \tag{56}$$

which indicates $q_k^p < q_k^z$. Similarly, we have $q_1^p > q_1^z$. Hence, $m = \max\{i | q_i^p \geq q_i^z, i = 1, 2, \ldots, k\}$ exists and $m < 1$. Using Equation (52), we have

$$\begin{cases} q_i^p \geq q_i^z & \text{if } 1 \leq i \leq m, \\ q_i^p < q_i^z & \text{if } m + 1 \leq i \leq k. \end{cases} \tag{57}$$

Directly applying $\sum_{i=1}^{k} q_i^z = \sum_{i=1}^{k} q_i^p = 1$ gives that

$$\sum_{i=1}^{m} \underbrace{q_i^p - q_i^z}_{\geq 0} = \sum_{i=m+1}^{k} \underbrace{q_i^z - q_i^p}_{>0}. \tag{58}$$

According to Lemma 4 and Equation (58), if we transport the redundancy from right to left, the entropy of the distribution will decrease. The transportation process is described as following:

Step 1 Let $i \leftarrow 1$ and $j \leftarrow k$.

Step 2 If $\Delta = \min\{q_i^p - q_i^z, q_j^z - q_j^p\} > 0$, then $q_i^z \leftarrow q_i^z + \Delta$ and $q_j^z \leftarrow q_j^z - \Delta$.

Step 3 If $q_i^p = q_i^z$, $i \leftarrow i + 1$. Else $j \leftarrow j - 1$.

Step 4 If $i \geq j$ , we finish this process. Else, we return step 2.

After at most $k - 1$ loops, $(q_1^z, q_2^z, \ldots, q_k^z)$ becomes $(q_1^p, q_2^p, \ldots, q_k^p)$ . Equation (58) ensures the correctness of the transportation process. According to Lemma 4, we have

$$H(q_1^z, q_2^z, \ldots, q_k^z) > H(q_1^p, q_2^p, \ldots, q_k^p), \tag{59}$$

which implies that $\text{erank}(\mathbb{C}_p) < \text{erank}(\mathbb{C}_z)$.

$\square$

### A.7 Proof of Theorem 4

*Proof.* Since $p_x^{(t+1)} = p_x^{(t)} - \alpha(p_x^{(t)} - z_{x+}^{(t)}) = (1 - \alpha)p_x^{(t)} + \alpha z_{x+}^{(t)}$ and $p_x = W z_x$,

$$\mathbb{C}_p^{(t+1)} = \mathbb{E}_x p_x^{(t+1)} (p_x^{(t+1)})^\top \tag{60}$$

$$= \mathbb{E}_{x,x+} \left( (1 - \alpha)p_x^{(t)} + \alpha z_{x+}^{(t)} \right) \left( (1 - \alpha)p_x^{(t)} + \alpha z_{x+}^{(t)} \right)^\top \tag{61}$$

$$= \mathbb{E}_x (1 - \alpha)^2 p_x^{(t)} \left( p_x^{(t)} \right)^\top + \mathbb{E}_{x+} \alpha^2 z_{x+}^{(t)} \left( z_{x+}^{(t)} \right)^\top$$

$$+ \mathbb{E}_{x,x+} \alpha(1 - \alpha) \left[ p_x^{(t)} \left( z_{x+}^{(t)} \right)^\top + z_{x+}^{(t)} \left( p_x^{(t)} \right)^\top \right] \tag{62}$$

$$= (1 - \alpha)^2 \mathbb{E}_x p_x^{(t)} \left( p_x^{(t)} \right)^\top + \alpha^2 \mathbb{E}_{x+} z_{x+}^{(t)} \left( z_{x+}^{(t)} \right)^\top$$

$$+ \alpha(1 - \alpha) \left[ W \mathbb{E}_{x,x+} z_x^{(t)} \left( z_{x+}^{(t)} \right)^\top + \mathbb{E}_{x,x+} z_{x+}^{(t)} \left( z_x^{(t)} \right)^\top W^\top \right] \tag{63}$$

$$= (1 - \alpha)^2 \mathbb{C}_p^{(t)} + \alpha^2 \mathbb{C}_z^{(t)} + \alpha(1 - \alpha) \left( W \mathbb{C}_+ + \mathbb{C}_+ W^\top \right). \tag{64}$$

It follows that

$$V \Lambda_p^{(t+1)} V^\top = (1 - \alpha)^2 V \Lambda_p^{(t)} V^\top + \alpha^2 V \Lambda_z^{(t)} V^\top + 2\alpha(1 - \alpha) V \Lambda_w^{(t)} \Lambda_+^{(t)} V^\top. \tag{65}$$

As a result, we have

$$\Lambda_p^{(t+1)} = (1 - \alpha)^2 \Lambda_p^{(t)} + \alpha^2 \Lambda_z^{(t)} + 2\alpha(1 - \alpha) \Lambda_w^{(t)} \Lambda_+^{(t)}, \tag{66}$$

*i.e.,*

$$\lambda_{i,t+1}^p = (1 - \alpha)^2 \lambda_{i,t}^p + \alpha^2 \lambda_{i,t}^z + 2\alpha(1 - \alpha) \lambda_{i,t}^w \lambda_{i,t}^+. \tag{67}$$

Dividing both sides of Equation (67) by $\lambda_{i,t}^p$, we have

$$\frac{\lambda_{i,t+1}^p}{\lambda_{i,t}^p} = (1-\alpha)^2 + \alpha^2 \frac{\lambda_{i,t}^z}{\lambda_{i,t}^p} + 2\alpha(1-\alpha)\frac{\lambda_{i,t}^w \lambda_{i,t}^+}{\lambda_{i,t}^p} \tag{68}$$

$$= (1-\alpha)^2 + \alpha^2 h_t^2(\lambda_{i,t}^p) + 2\alpha(1-\alpha)h_t(\lambda_{i,t}^p)\frac{\lambda_{i,t}^+}{\lambda_{i,t}^z}, \tag{69}$$

where $h_t(\lambda_{i,t}^p) = 1/\lambda_{i,t}^w = \sqrt{\frac{\lambda_{i,t}^z}{\lambda_{i,t}^p}}$.

- If $W = W^* = \mathbb{C}_+^{(t)}(\mathbb{C}_z^{(t)})^{-1}$, then $h_t(\lambda_{i,t}^p) = \frac{\lambda_{i,t}^z}{\lambda_{i,t}^+}$. Hence, $\frac{\lambda_{i,t+1}^p}{\lambda_{i,t}^p} = 1 - \alpha^2 + \alpha^2 h_t^2(\lambda_{i,t}^p)$ is monotonically decreasing and non-constant.

- If $\lambda_{i,t}^z = \lambda_{i,t}^+$, then $\frac{\lambda_{i,t+1}^p}{\lambda_{i,t}^p} = (1-\alpha)^2 + \alpha^2 h_t^2(\lambda_{i,t}^p) + 2\alpha(1-\alpha)h_t(\lambda_{i,t}^p)$ is monotonically decreasing and non-constant ($0 < \alpha < 1$).

According to Theorem 3, we have $\text{erank}(\mathbb{C}_{p^{(t+1)}}) > \text{erank}(\mathbb{C}_{p^{(t)}})$.

$\square$

## B EXPERIMENTAL DETAILS

In this section, we provide the details and hyperparameters for SymSimSiam and variants of spectral filters.

### B.1 DATASETS

We evaluate the performance of our methods on four benchmark datasets: CIFAR-10, CIFAR-100 (Krizhevsky et al., 2009), ImageNet-100 and ImageNet-1k (Deng et al., 2009). CIFAR-10 and CIFAR-100 are small-scale datasets, composed of $32 \times 32$ images with 10 and 100 classes, respectively. ImageNet-100 is a subset of ImageNet-1k containing 100 classes.

### B.2 IMPLEMENTATION DETAILS

Unless specified otherwise, we follow the default settings in solo-learn (da Costa et al., 2022) on CIFAR-10, CIFAR-100 and ImageNet-100. For ImageNet, our implementation follows the official code of SimSiam (Chen & He, 2021), and we use the same settings. For a fair comparison, SimSiam with a learnable linear predictor is adopted as our baseline. With the original projector, SimSiam with a learnable linear predictor could not work, so we delete the last BN in the projector in this case. And we also list the results of SimSiam with the default predictor (we refer it as a learnable nonlinear predictor).

**Data augmentations.** The augmentation pipeline is RandomResizedCrop with scale in (0.2, 1.0), RandomHorizontalFlip with probability 0.5, ColorJitter (brightness (0.4), contrast (0.4), saturation (0.4), hue (0.1)) with probability 0.8 and RandomGray with probability 0.2. For ImageNet-100 and ImageNet-1k, Gaussian blurring (Chen et al., 2020) with an applying probability 0.5 is also used.

**Optimization.** SGD is used as the optimizer with momentum 0.9 and weight decay $1.0 \times 10^{-5}$ ($1.0 \times 10^{-4}$ for ImageNet-1k). The learning rate adopts the linear scaling rule (lr×BatchSize/256) with a base learning rate of 0.5 (0.05 for ImageNet-1k). After 10 epochs of warmup training, we use the cosine learning rate decay (Loshchilov & Hutter, 2017). We use a batch size 256 on CIFAR-10 and CIFAR-100; 128 on ImageNet-100 and 512 on ImageNet-1k.

**Linear evaluation.** For the linear evaluation, we evaluate the pre-trained backbone network by training a linear classifier on the frozen representation. For CIFAR-10, CIFAR-100 and ImageNet-100, the linear classifier is trained using SGD optimizer with momentum = 0.9, batch size = 256 and initial learning rate = 30.0 for 100 epochs. The learning rate is divided by 10 at epochs 60 and 80. For ImageNet-1k, following the official code of SimSiam (Chen & He, 2021), we train the linear classifier for 90 epochs with a LARS optimizer (You et al., 2017) with momentum = 0.9, batch size = 4096, weight decay = 0, initial learning rate = 0.1 and cosine decay of the learning rate.

### B.3 Experimental Setting for Figures

For the implementations of BYOL, SimSiam, SwAV and DINO on CIFAR-10 and CIFAR-100, we use codes and all default settings in the open-source library solo-learn (da Costa et al., 2022). For SwAV, we do not store features from the previous batches to augment batch features (queue_size = 0) for the consistency of training loss. We adopt ResNet-18 as the backbone encoder. The dimensions of the outputs of BYOL, SimSiam, SwAV and DINO are 256, 2048, 3000 and 4092, respectively.

In Figure 3, we use the alignment metric to measure the eigenspace alignment of the online and target outputs, i.e., $\mathbb{C}_p$ and $\mathbb{C}_z$. The intuition is that, given $u_i$ as an eigenvector of $\mathbb{C}_z$, if it is also an eigenvector of $\mathbb{C}_p$, then $\mathbb{C}_p u_i = \lambda_i' u_i$ is in the same direction as $u_i$. Thus, the higher the cosine similarity between $u_i$ and $\mathbb{C}_p u_i$, the more aligned between the eigenspace of $\mathbb{C}_p$ and $\mathbb{C}_z$. Formally, we define the alignment between the eigenspace of $\mathbb{C}_p$ and $\mathbb{C}_z$ as

$$Alignment(\mathbb{C}_p, \mathbb{C}_z) = \frac{1}{m} \sum_{i=1}^{m} \frac{u_i^T}{\|u_i\|_2} \frac{\mathbb{C}_p u_i}{\|\mathbb{C}_p u_i\|_2}, \tag{70}$$

where $u_i$ is eigenvector corresponding to the $i$-th largest eigenvalue $\lambda_i^z$ of $\mathbb{C}_z$, $i = 1, 2, \ldots, m$. And $m$ is set to 512 for SimSiam, SwAV and DINO and 128 for BYOL. The sum of the first $m$ eigenvalues is greater than 99.99% of the sum of all eigenvalues. Therefore, we can think that the space span by the first $m$ eigenvectors is a good approximation to the original eigenspace. And in Figures 4, 5 & 8, the first 512 point pairs are displayed for SimSiam, SwAV and DINO (128 for BYOL).

### B.4 Pseudocode

Here, we provide the pseudocode for SymSimSiam (Algorithm 1) and variants of spectral filters (Algorithm 2).

---

**Algorithm 1** SymSimSiam: Pseudocode in a PyTorch-like style.

---

```
# f: backbone + projection mlp
# c: center (1-by-k)
# m: momentum 0.9

for x in loader:                    # load a minibatch x with n samples
    x1, x2 = aug(x), aug(x)         # two random augmentations
    z1, z2 = f(x1), f(x2)           # projections, n-by-k
    z1 = normalize(z1, dim=1)       # l2-normalize
    z2 = normalize(z2, dim=1)       # l2-normalize

    update_center(cat(z1,z2))
    z1, z2 = z1-c, z2-c

    loss = -(z1*z2).sum(dim=1).mean()

    loss.backward()
    update(f.params)                # SGD update

@torch.no_grad()
def update_center(z):
    c = m * c + (1-m) * z.mean(dim=0)
```

---

## C Additional Results Based on BYOL Framework

In the main paper, we conduct experiments based on the SimSiam framework. Here, we also gather some results of variants of target high-pass filters based on the BYOL framework. We adopt ResNet-18 as the backbone encoder and use the default setting in da Costa et al. (2022).

Table 4 shows that our target high-pass filters can often outperform BYOL with a large margin at earlier epochs. In particular, on CIFAR-10, the target filter $h(\sigma) = \sigma^{-0.3}$ improves BYOL with learnable linear predictor by 4.92%, 2.68% and 1.00% with 100, 200 and 400 epochs, respectively.

**Algorithm 2** Variants of spectral filters: Pseudocode in a PyTorch-like style.

```
# f: backbone + projection mlp
# g: spectral filter
# location: the location of spectral filter (online or target)

for x in loader:                    # load a minibatch x with n samples
    x1, x2 = aug(x), aug(x)         # two random augmentations
    z1, z2 = f(x1), f(x2)           # projections, n-by-k

    loss = L(z1,z2,location)/2 + L(z2,z1,location)/2

    loss.backward()
    update(f.params)                # SGD update

def L(p,z,location):                # loss function
    if location == "online":        # online predictor
        with torch.no_grad():
            u, s, vh = svd(p)
            w = vh.T @ diag(g(s)) @ vh
        p = p @ w.detach()
    else:                           # target predictor
        with torch.no_grad():
            u, s, vh = svd(z)
            z = u @ diag(g1(s)) @ vh   # g1(t) = tg(t)

    z = z.detach()                  # stop gradient
    p = normalize(p, dim=1)         # l2-normalize
    z = normalize(z, dim=1)         # l2-normalize
    return -(p*z).sum(dim=1).mean()
```

Table 4: Linear probing accuracy (%) of BYOL with different target predictors on CIFAR-10/100.

| | CIFAR10 | | | CIFAR100 | | |
|---|---|---|---|---|---|---|
| Predictor | 100ep | 200ep | 400ep | 100ep | 200ep | 400ep |
| BYOL (nonlinear predictor) | 82.14 | 87.89 | 91.24 | 53.92 | 60.75 | 67.41 |
| BYOL (linear predictor) | 80.81 | 86.58 | 89.73 | 49.65 | 57.53 | 60.66 |
| $h(\sigma) = \sigma^{-0.3}$ | **85.73** | **89.26** | **90.73** | 57.56 | 61.89 | **65.88** |
| $h(\sigma) = \sigma^{-0.5}$ | 85.46 | 89.24 | 90.30 | 58.12 | 62.51 | 65.16 |
| $h(\sigma) = \sigma^{-0.7}$ | 85.08 | 88.21 | 90.01 | 58.20 | **62.58** | 65.72 |
| $h(\sigma) = \sigma^{-1}$ | 84.64 | 88.29 | 89.96 | **58.40** | 62.06 | 65.04 |

On CIFAR-100, the filter $h(\sigma) = \sigma^{-1}$ is better 8.75% than the baseline with 100 epochs, the filter $h(\sigma) = \sigma^{-0.7}$ improves the baseline by 5.05% with 200 epochs and $h(\sigma) = \sigma^{-0.3}$ improves the baseline by 5.22% with 400 epochs.

## D  COST COMPARISON

We compare the training speeds and GPU memory usages of different methods (SimSiam, the online filter $g(\sigma) = \log(1 + \sigma)$ and the target filter $h(\sigma) = \sigma^{-0.3}$) on CIFAR-10. We perform our experiments on a single RTX 2080ti GPU.

As shown in Table 5, compared to the original SimSiam, the GPU memory usage of our methods only increase a little (26 MiB). SimSiam and the target filter $h(\sigma) = \sigma^{-0.3}$ have the same training time for 20 epochs.

Table 5: Training time and memory comparison of different methods on CIFAR-10.

| Method | Total time for 20 epochs | GPU memory |
|---|---|---|
| SimSiam | 527 s | 3043 MiB |
| $g(\sigma) = \log(1 + \sigma)$ | 532 s | 3069 MiB |
| $h(\sigma) = \sigma^{-0.3}$ | 527 s | 3069 MiB |

Table 6: Eigenspace alignment between $\mathbb{C}_z$ and $\mathbb{C}_+$.

| Method | BYOL | SimSiam | SwAV | DINO |
|---|---|---|---|---|
| CIFAR-10 | 0.9897 | 0.9979 | 0.9965 | 0.9734 |
| CIFAR-100 | 0.9892 | 0.9972 | 0.9956 | 0.9621 |

# E ADDITIONAL VISUALIZATION

## E.1 ADDITIONAL EMPIRICAL EVIDENCE FOR EIGENSPACE ALIGNMENT

In the main paper, we have shown that $\mathbb{C}_p$ and $\mathbb{C}_z$ share the same eigenspace. Here, we add empirical evidence for eigenspace alignment between $\mathbb{C}_z$ and $\mathbb{C}_+$ on CIFAR-10 and CIFAR-100. As shown in Table 6, we can see that all methods have very high alignment between the eigenspace of $\mathbb{C}_z$ and $\mathbb{C}_+$.

## E.2 RESULTS ON CIFAR-100

We conduct some experiments on CIFAR-100. Figure 7 demonstrates that the target branch always has higher effective rank than the online branch and the rank of the online branch continues to increase after the warmup state in all methods.

In Figure 8(a), we compare the eigenvalues computed from two branch outputs. There is a clear difference in eigenvalues, and the eigenvalue distribution of the target branch is more biased towards larger values. Figure 8(b) shows the spectral filter $g(\lambda_i^z) = \sqrt{\lambda_i^p / \lambda_i^z}$, where $\lambda_i^p, \lambda_i^z$ are eigenvalues of online and target correlation matrices $\mathbb{C}_p, \mathbb{C}_z$, respectively. The spectral filters of all methods are nearly monotonically increasing *w.r.t.* $\lambda_i^z$.

## E.3 TRAINING DYNAMIC

In Figure 9, we compare the training loss between SimSiam and SymSimSiam on CIFAR-10 and CIFAR-100. We can see that the loss of SimSiam is always larger than SymSimsiam and does not consistently decrease. Instead, the alignment loss of SymSimSiam continues to decrease.

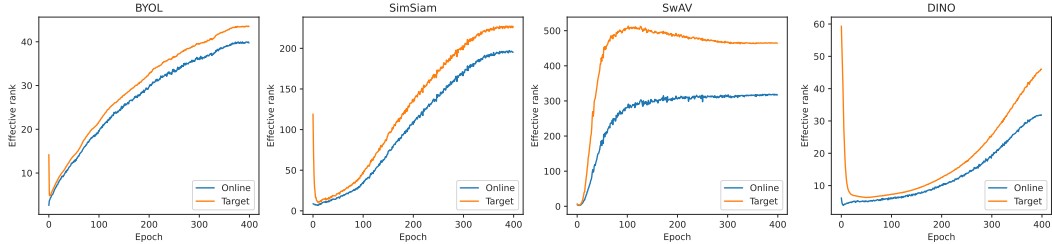

Figure 7: The effective rank of the normalized outputs of the online and target branches along the training dynamics on CIFAR-100.

# F COMPARISON TO RELATED WORK

**Comparison with Tian et al. (2021)**. Although we assume eigenspace alignment as in Tian *et al.* , we take very different techniques and arrive at different conclusions, as highlighted below:

- **Difference Perspectives, Techniques, and Conclusions**. As for the goal, Tian *et al.* only consider how predictor helps avoid full collapse. Instead, we first point out that avoiding full collapse is not the key role of asymmetric designs (also achievable with symmetric designs). Thus, we focus on the more general dimensional collapse issue and analyze this

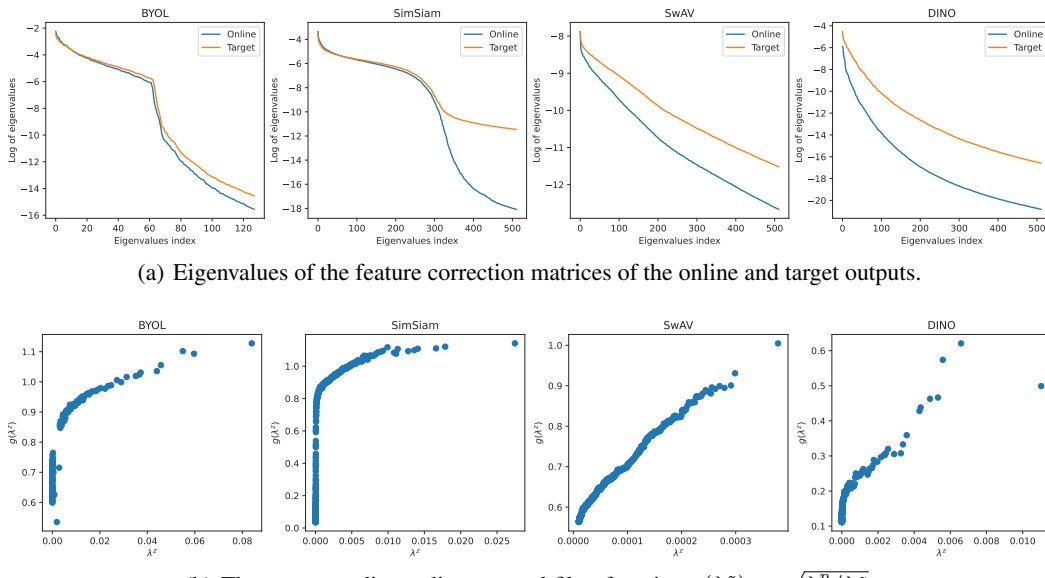

(a) Eigenvalues of the feature correction matrices of the online and target outputs.

(b) The corresponding online spectral filter function $g(\lambda_i^z) = \sqrt{\lambda_i^p/\lambda_i^z}$.

Figure 8: Rank difference experiments and spectral filters on CIFAR-100.

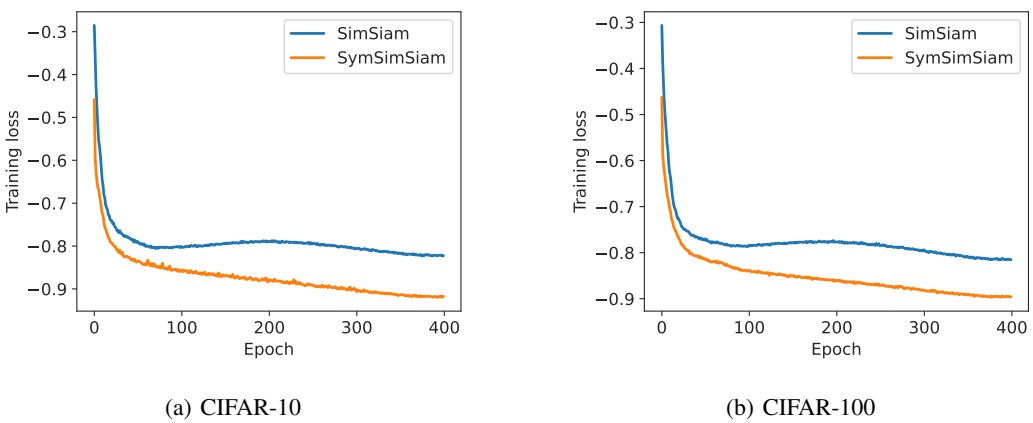

(a) CIFAR-10

(b) CIFAR-100

Figure 9: Training loss of SimSiam and SymSimSiam on CIFAR-10 and CIFAR-100.

quantitatively through the change of the effective rank along training. As for the techniques, Tian *et al.* mainly analyze the linear learning dynamics under strong architectural and data assumptions, while ours focus on the common spectral filtering property that also holds for nonlinear modules and general data distributions. As for the conclusion, we formally show that asymmetric designs will improve effective dimensionality, while Tian *et al.* only discuss how it avoids full collapse (which is an extreme case of dimensional collapse, and a non-full-collapse encoder may still suffer from dimensional collapse).

- **A unified framework for various asymmetric designs**. Tian *et al.* 's analysis only focuses on the predictor in BYOL and SimSiam, and they cannot explain why SwAV and DINO also work without predictors. Our RDM applies to all existing asymmetric designs through the unified spectral filter perspective.

- **General principles for predictor design.** Tian *et al.* propose DirectPred, which is only a specific filter. Instead, we point out the core underlying principle, that as long as the online filter is a low-pass filter, it could theoretically avoid feature collapse. We also empirically verify this point by showing that various online low-pass filters can avoid feature collapse.

- **A More Effective Asymmetric Design through Target Predictor**. Based on our RDM theory, we also propose a new kind of asymmetric design in non-contrastive learning: applying a predictor to the target branch. We show that target predictors achieve better results than online predictors while being more computationally efficient.

Therefore, our analysis improves Tian et al. (2021) in many aspects and apply to a wider context. And we achieve this with new perspectives and techniques that are quite distinctive from Tian et al. (2021).

