# OpenReview forum: "Towards a Unified Theoretical Understanding of Non-contrastive Learning via Rank Differential Mechanism"
_ICLR.cc/2023/Conference — ICLR 2023 poster_

### Official Review · Reviewer_rcoc · 2022-10-24

**Confidence:** 4
**Correctness:** 2
**Technical Novelty And Significance:** 4
**Empirical Novelty And Significance:** 2
**Recommendation:** 6

**Clarity, Quality, Novelty And Reproducibility:**

The paper is mostly clearly written, besides some issues with the presentation of the theoretical results that are described above. The findings are novel, to the best of my knowledge.

**Strength And Weaknesses:**

**Strengths**

The paper makes many interesting observations about non-contrastive learning, a promising alternative to contrastive learning that is more computationally efficient and scalable.
- Symmetric architectures, with feature centering, are shown to avoid complete collapse but not dimension collapse
- The existence of rank differential in many non-contrastive methods is a new and interesting finding in my opinion
- The overall story, with eigenspace alignment -> rank difference -> rank improvement seems plausible given the results of the paper. Many claims are supported either by some experiments or some theory, albeit with many assumptions at times to make analysis more convenient.
  - Eigenspace alignment is empirically verified
  - Rank difference is empirically verified and justified from eigenspace alignment by further assuming a “low-pass filter” assumption (that is empirically verified)
  - Finally Theorem 4 proves rank improvement as a consequence of the above two (potentially significant issues with this in weaknesses).
- The rank differential and low/high pass filtering view inspire new non-contrastive variants that directly set the prediction head rather than training them (similar to DirectPred). These methods perform comparably to previous methods, sometimes outperforming them with fewer epochs.

Overall many interesting insights are provided which can help understand non-contrastive methods better.


**Weaknesses**

- My main concern is with Theorem 4 that tries to explain why eigenspace alignment + rank differential can lead to increase in effective rank. In particular, I'm not sure that the result is correct, due to a potential bug in Eq. 22 in the Appendix. While it is assumed that the (unnormalized) covariances $\Gamma_{t}$ and $\Gamma_{z}$ of $p$ and $z$ respectively are aligned, that does not imply that $\mathbb{E} [p^{\top} z] = \left(\Gamma_{p} \Gamma_{z}\right)^{1/2}$, as assumed in Eq. 22. (Simple counter example is when $p$ and $z$ are independent, and zero-mean Gaussians say). For the result to hold, you might something stronger that the cross-covariance of $p$ and $z$ are also aligned, which is a much stronger assumption and will change the expression for Eq 4. and Theorem 4 in general.

- Most of the theoretical results are not end-to-end (which is fine) and make many assumptions as the paper goes along, which is sometimes hard to follow. For e.g. why does the assumption in Theorem 5 that $z$ and $\bar{z}$ have aligned eigenspaces makes sense?

Other comments/questions:

- While Theorem 4 is meant to show that the effective rank keeps increasing, what stops the rank to asymptotically stagnate at a small value and lead to dimension collapse? Furthermore, is there any intuition for why rank differential should always remain high as training progresses (besides empirical evidence)?

- The use of the terms “low/high pass filters” can be confusing at times. If these terms are directly inspired from signal processing (or another field), then the connection should be made clearer. The notation for $g$ and $h$ are also ambiguous, since they are just functions of a single $\lambda^{z}\_{i}$ but their definitions also involve $\lambda^{p}\_{i}$.

- Figure 4: typo “online branch is more biased towards larger values”. For SimSiam, plot (a) suggests that the ratio of $\lambda^{p}\_{i} / \lambda^{z}\_{i}$ is not monotonic, but figure (b) doesn’t agree since the ratio seems to be 1 at the ends. Do the spectral filter function plots not include all indices $i$? Is this also true for other plots like Figures 5 and 6?

**Summary Of The Paper:**

This paper studies how representation collapse is avoided in non-contrastive methods that employ certain “asymmetrical tricks” rather than using negative samples, despite the existence of trivial solutions.
It proposes an explanation through an idea of “rank differential” created by asymmetric designs, i.e. the effective rank of representations from the target branch is consistently higher than that of the online branch, as training progresses.
This is empirically verified for methods like BYOL, SimSiam, SwAV, DINO.
The paper hypothesizes that this rank differential leads to a continuous increase in the effective rank of representation and thus prevents dimensional collapse; a combination of theory and experiments are used to justify this.
The paper further attempts to explain the existence of such a rank differential through a “low-pass filtering” effect of asymmetric designs on the representation spectrum.
Motivated by these ideas, new non-contrastive variants (that directly set the prediction head) are proposed that achieve comparable performance to existing methods on CIFAR and ImageNet benchmarks.

**Summary Of The Review:**

Overall I believe that the paper contributes many interesting insights about non-contrastive methods through the lens of rank differences, and also suggest ways to empirically leverage these findings. However currently I am not convinced about the technical correctness and conclusions about one of the main results (Theorem 4) that aims to prove why rank differential can help avoid dimension collapse, which is one of the main contributions of the paper. Keeping this and some other issues in mind, I am inclined to assign a score of weak reject (keeping in mind all the other contributions).

---

> ### Author Response · Authors · 2022-11-14
> **Response to Reviewer rcoc (2/2)**
>
>
> **Q4.** The use of the terms “low/high pass filters” can be confusing at times. If these terms are directly inspired from signal processing (or another field), then the connection should be made clearer. The notation for $g$ and $h$ are also ambiguous, since they are just functions of a single $\lambda_i^z$ but their definitions also involve $\lambda_i^p$.
>
> **A4**. Thanks for your suggestions. We have added introductions of spectral filters in Sec 4.1. We have also revised the definition of $g$ and $h$ to be more clear on the definition.
>
> ---
>
> **Q5.**  Figure 4: typo “online branch is more biased towards larger values”. For SimSiam, plot (a) suggests that the ratio of $λ_i^p/λ_i^z$ is not monotonic, but figure (b) doesn’t agree since the ratio seems to be 1 at the ends. Do the spectral filter function plots not include all indices i? Is this also true for other plots like Figures 5 and 6?
>
> **A5.**  Thanks for your comments, we have updated Figures 4 & 5 and the caption of Figures 4. The detailed experiment setting for Figures 4 & 5 can be found in Appendix B.3, and we have now mentioned some in the captions of Figures 4 & 5. In Figure 4 (both (a) and (b)), top $m$ eigenvalues (indexes) are shown (128 for BYOL and 512 for others) and the sum of the first $m$ eigenvalues is greater than 99.99 % of the sum of all eigenvalues. Figures 5(a) & (b) are plotted similarly. As for Figure 6, we plotted the eigenvalues which are bigger than $e^{-25}$ as Figure 2 in [1].
>
> [1] Li Jing, Pascal Vincent, Yann LeCun, and Yuandong Tian. Understanding dimensional collapse in contrastive self-supervised learning. ICLR 2022. [https://openreview.net/pdf?id=YevsQ05DEN7](https://openreview.net/pdf?id=YevsQ05DEN7)
>
> ---
>
> We thank Reviewer rcoc again for your careful reading and constructive comments. Please let us know if there is more to clarify. We are happy to take more questions before the rebuttal stage ends. Looking forward to your reply!

---

> ### Author Response · Authors · 2022-11-14
> **Response to Reviewer rcoc (1/2)**
>
> We thank Reviewer rcoc for your careful reading and for appreciating the insights and effectiveness of our theory. Below, we address your concerns point by point.
>
> ---
>
> **Q1.** Concerns on Theorem 4  due to a potential bug in Eq. 22 in the Appendix.
>
> **A1**. Thanks for pointing it out, and we have fixed it in our revision. Indeed, we cannot conclude $E_xp_xz_x^\top$ has the same eigenspace as $C_p$ or $C_z$ without additional assumptions. To mitigate this issue, in the revision, we consider the equivalent spectral filter $p_x=W_gz_x$ (**Lemma 2**) in the derivation. Then we can derive that $E_xp_xz_x^\top=E_xWz_xz_x^\top=WC_z$, whose eigenvalues are $\sqrt{\lambda^p/\lambda^z}\cdot\lambda^z=\sqrt{\lambda^p\lambda^z}$. To be more rigorous, we further take the alignment between the positive pairs $z_x,z_{x^+}$ in to consideration, add theoretical justifications for alignment (**Lemma 1**), and revise Mechanism 1 and Theorem 4 accordingly. **After the revision above, the proof of Theorem 5 is now complete. Hope it could address your concerns.**
>
> ---
>
> **Q2**.  Most of the theoretical results are not end-to-end (which is fine) and make many assumptions as the paper goes along, which is sometimes hard to follow.
>
> **A2**. To better present and justify our theory, we have reorganized this paper (particularly Section 4), to provide better theoretical justifications for the proposed mechanism. ***After revision, each mechanism is now supported with both theoretical justification and empirical verification***. Below, we further address the specific question that is unclear to you.
>
> > Why does the assumption in Theorem 5 that $z$ and $\bar z$ have aligned eigenspaces makes sense?
> >
>
> For clarity, we further add a theoretical justification for this property in **Lemma 1 (new),** and empirically verify it on real-world data in **Appendix E.1**. Here we also provide an intuitive understanding of this phenomenon. According to **Lemma 3 (new)**, the correlation matrix of $\bar z$, $C_{\bar z}=E_{\bar x}z_{\bar x}z_{\bar x}^\top$, is equivalent to the correlation between two positive views $C_+=E_{x,x^+}z_xz_{x^+}^\top$, i.e., $C_{\bar z}=C_+$.
>
> After learning with alignment loss, the two positive views will have very similar representations, i.e., $z_x$ is approximately $z^+_x$.
>
> Therefore, we have  $C_+= E_{x,x^+}z_xz_{x^+}^\top\approx E_{x,x^+}z_xz_{x}^\top=C_z$. Combining the two results above, we can see that $\bar z$ and $z$ indeed have similar correlation, i.e., $C_{\bar z}=C_+\approx C_z$, which implies that they will have aligned eigenspace.
>
> ---
>
> **Q3**. Two questions about Theorem 4.
>
> **A3.** We address your concerns point by point.
>
> > **Point a**. While Theorem 4 is meant to show that the effective rank keeps increasing, what stops the rank to asymptotically stagnate at a small value and lead to dimension collapse?
> >
>
> We are afraid that we do not fully understand your point here. Fig 1 has shown that the rank of the online target keeps increasing to large values along training in all existing non-contrastive methods. Perhaps, you are asking why fully symmetric methods, like SymSimSiam, will not have such a rank-lifting process. In fact, we have discussed this phenomenon in Sec 4.2 (below Lemma 2). The main reason is that the fully symmetric architecture cannot create a rank difference between two symmetric branches. Without the rank difference caused by the low-pass filtering effect of asymmetric modules (the condition in Theorem 4), the alignment process will no longer improve the effective rank as in Theorem 4. Please let us know if there is more to clarify here.
>
> > **Point b**. Furthermore, is there any intuition for why rank differential should always remain high as training progresses (besides empirical evidence)?
> >
>
> We actually do not need the rank difference to be “high”. Actually, as long as there is a **consistent** rank difference (target > online), it could alleviate dimensional collapse. As shown in Fig 1, the differences between target and online ranks are not that large on BYOL and SimSiam, but it is **still enough to lift the rank of the online output**. Intuitively, as long as target > online, aligning the online output to the (detached) target output will enforce the online feature to strive for a larger rank in order to match the target.

---

> ### Comment · Reviewer_rcoc · 2022-11-19
> **Response to authors**
>
> Thank you for the detailed response and the revision. I haven’t had the chance to thoroughly go through all of the changes, but on a rough read of the response and the changes, I believe that my main concerns about correctness of the results have been addressed. I will be increasing my score to at least a weak accept in light of this.
>
> A couple of follow-up comments:
>
> - Under the new setting, it is unclear why $\tilde{p}$ and Lemma 2 are needed. Some clarification in the paper would be helpful.
>
> - Regarding A3a, one could have a monotonically increasing function that is upper bounded, e.g. sigmoid. So my question was about what guarantees that effective rank does not asymptotically saturate to a suboptimal value, leading to dimension collapse.

---

> > ### Author Response · Authors · 2022-11-22
> > **Thanks and Further Response**
> >
> > Thanks for your reply and for appreciating our response!  We address your further comments below.
> >
> > ---
> >
> > **Q1.** Under the new setting, it is unclear why $\tilde {p}$ and Lemma 2 are needed. Some clarification in the paper would be helpful.
> >
> > **A1.** This is mainly because for general cases, such as a nonlinear predictor $g$, i.e., $p=g(z)$, it is hard to obtain a closed-form solution for $E_x p_xz_x^\top$due to their complex nonlinear dependency. Instead, **Lemma 2** shows that under eigenspace alignment (verified in practice), **a nonlinear predictor $p=g(z)$ is nearly equivalent to a linear transformation $p=W_gz$ (specifically, a spectral filter)** in the sense that they have the same feature correlation. Therefore, we use the linear spectral filter instead for a tractable  analysis of the change of eigenvalues and effective rank during the gradient update.
> >
> > Thanks for your suggestions, and we will add the explanation to the paper in future updates.
> >
> > ---
> >
> > **Q2**. Regarding A3a, one could have a monotonically increasing function that is upper bounded, e.g. sigmoid. So my question was about what guarantees that effective rank does not asymptotically saturate to a suboptimal value, leading to dimension collapse.
> >
> > **A2**. Thanks for your explanation and we now understand your point. Indeed, the current analysis mainly tells us that in general cases, the effective rank (erank) will increase, but not specifically how much it will increase. In fact, in **Fig 1**, we observe that although erank keeps increasing along training, **their eranks still saturate at some medium points** (different values for different methods), not attaining an exact uniform distribution (with maximal erank) at last. Therefore, it is even hard to tell **whether these methods themselves could attain the “optimal value”**. One thing we can be sure, though, is that as shown in **Fig 2c**, these asymmetric designs can indeed help lift the erank significantly compared to symmetric ones and increase the feature diversity. Here, we provide a preliminary theoretical explanation for why the asymmetric alignment could help escape from a severe collapse.
> >
> > In the following proposition, we show that if there is a large gap between the largest and smallest eigenvalues (implying large $q_1/q_k$), which corresponds to a severe collapse, then the alignment update will induce a large increase in erank that is polynomial in the ratio $q_1/q_k$. Therefore, we can conclude that **a severe collapse (a large gap) will also induce a very large update of effective rank**, **which will prevent the features from saturating at too collapsed solutions**.
> >
> > > **Proposition 1.**  Let $q_i^{(t)} = \frac{\lambda_{i,t}^p}{\sum_j\lambda_{j,t}^p}$ and $q^{(t)}= (q^{(t)}_1, \dots, q^{(t)}_k)$,
> >
> > > where $\lambda_{i,t}^p$ is $i$-th eigenvalue of the online correlation $C_p^{(t)}$ at step $t$. Assume $q_1\geq q_2\dots\geq q_k\geq\mu$, and denote $\frac{q_1^{(t+1)}}{q_1^{(t)}} = 1-\epsilon_1$ and $\frac{q_k^{(t+1)}}{q_k^{(t)}} = 1+\epsilon_2$ ($\epsilon_1, \epsilon_2 >0$ according to our proof of Theorem 3).
> >
> > > Then, we have $\operatorname{erank}(C_p^{(t+1)}) > (\frac{q_1^{(t)}}{q_k^{(t)}})^{\Delta}\operatorname{erank}(C_p^{(t)})$, where $\Delta = \min(\{\epsilon_1q_1^{(t)} ,\epsilon_2q_k^{(t)}\}) \geq \mu \min(\{\epsilon_1 ,\epsilon_2\})$.
> >
> > We will add this result and its proof in future paper updates. And we will keep exploring more quantitative characterization of the convergence of erank in the future.
> >
> > ---
> >
> > Thanks for your response and hope our new explanations above could ease your concerns. Please let us know if there is more to clarify. We are willing to take you additional questions during the rebuttal stage.

---

### Official Review · Reviewer_xJ6R · 2022-10-25

**Confidence:** 5
**Correctness:** 4
**Technical Novelty And Significance:** 4
**Empirical Novelty And Significance:** 4
**Recommendation:** 6

**Clarity, Quality, Novelty And Reproducibility:**

The clarity is good. I can easily understand this paper, including proofs. The quality is also ok for me. The novelty and originality of the work are the advantages of this work. They focus on a very important topic: dimensional collapse in contrastive learning, and show us something new for this.

**Strength And Weaknesses:**

Strength
1. This paper proposes a novel perspective for non-contrastive learning methods. This perspective can effectively explain the four main approaches in non-contrastive learning.
2. Some of the more interesting variants were obtained under the new view. These methods are promising and are able to inspire the community.
3. The experiments are convincing in my opinion.

Weakness:
1. Although the authors call the method rank difference, and define an effective rank difference to estimate how uniform the eigenvalues are. It is the eigenvalues difference rather than the rank difference. I don't think that's a suitable name.
2. The eigenspace alignment assumption is a little bit strong, and all conclusions and methods are based on that. This paper is, therefore, more like an extension of ICML2021(Understanding self-supervised learning dynamics without contrastive pairs).

**Summary Of The Paper:**

In this paper, authors provide a new perspective for non-contrastive learning methods. A rank difference mechanism (RDM) theory is proposed to explain how existing non-constrastive learning methods alleviate dimensional feature collapse. More importantly,  RDM theory also provides practical guidelines for designing many new non-contrastive variants. In this paper, authors demonstrate the low-pass filters and high-pass filters for online and target branches respectively. In the experiments, the theory is consistent with the practical settings.

**Summary Of The Review:**

Based on the comments above, I am inclined to accept this paper. However, I will also take full account of other reviewers' comments.

---

> ### Author Response · Authors · 2022-11-14
> **Response to Reviewer xJ6R**
>
> We thank Reviewer xJ6R very much for appreciating the novelty of our analysis, and the effectiveness of our proposed methods. We address your concerns below.
>
> ---
>
> **Q1**. Why is this method called rank difference instead of eigenvalue difference?
>
> **A1**. We understand your concerns that our technical discussions mainly deal with the eigenvalues instead of the rank. From our general perspective, eigenvalue and rank are views of the same thing from two different levels: **eigenvalues from a micro level, while rank from a macro level**. Specifically, effective rank describes the overall distribution of the eigenvalues, and it determines the overall degree of feature collapse. Remarkably, the key inspiration of our theory is Fig 1, where we observe that target features have **a consistently higher rank** than the online features. This phenomenon provides an intuitive understanding of non-contrastive methods: **as long as this consistent rank difference exists, the target output will keep lifting the rank of the online output under their alignment loss**. The RDM theory is thus developed to provide theoretical justifications of **this rank-lifting process** by investigating the micro-level change of eigenvalues. Therefore, we believe that the macro-level rank concept could be a better grasp of the key insight of our theory. We would also like to hear more about your considerations if you would still prefer the eigenvalue name.
>
> ---
>
> **Q2**. The eigenspace alignment assumption is a little bit strong, and all conclusions and methods are based on that.
>
> **A2**. As observed in practice (Fig 3), there is indeed a strong alignment of eigenspace, which is almost always 1 in BYOL, SimSiam  and SwAV. On DINO, the alignment also converges to 1.  Therefore, the eigenspace alignment indeed holds on real-world data, making it a practical assumption. To further address your concerns, we added a new theoretical result, Lemma 1, where we show that under simplified linear settings (same as Tian et al), **our eigenspace alignment assumption indeed holds**. Therefore, we believe that both theoretical and empirical results have provided strong evidence for the alignment.
>
> ---
>
> **Q3**. This paper is, therefore, more like an extension of Tian et al (2021).
>
> **A3**. Although we generally assume eigenspace alignment as in Tian et al, we **take very different techniques and arrive at different conclusions**, as highlighted below:
>
> - **Difference Perspectives, Techniques, and Conclusions**.
>     - As for the goal, Tian et al only consider how predictor helps avoid full collapse. Instead, we first point out avoiding full collapse is **not the key role of asymmetric designs (also achievable with symmetric designs, Sec 3).** Thus, we focus on the **more general** **dimensional collapse** issue and analyze this **quantitatively** through the change of the **effective rank** along training.
>     - As for the techniques, Tian et al mainly analyze the linear **learning dynamics** under strong architectural and data assumptions, while ours focus on the **common** **spectral filtering** property that also holds for nonlinear modules and general data distributions.
>     - As for the conclusion, we formally show that asymmetric designs will improve effective dimensionality (Thm 4), while Tian et al only discuss how it avoids full collapse (which is an extreme case of dimensional collapse, and a non-full-collapse encoder may still suffer from dimensional collapse).
> - **A unified framework for various asymmetric designs**. Tian et al’s analysis only focus on the predictor in BYOL and SimSiam, and they cannot explain why SwAV and DINO also work without predictors. Our RDM applies to all existing asymmetric designs through the unified spectral filter perspective.
> - **General principles for predictor design**. Tian et al propose DirectPred, which is only a specific filter. Instead, we point out the core underlying principle, that as long as the online filter is a low-pass filter, it could theoretically avoid feature collapse (Thm 4). We also empirically verify this point by showing that various online low-pass filters can avoid feature collapse (Tab 1 & Fig 6).
> - **A More Effective Asymmetric Design through Target Predictor**. Based on our RDM theory, we also propose a **new kind of asymmetric design in non-contrastive learning: applying a predictor to the target branch.** We show that target predictors achieve better results than online predictors (Tab 1,2,3) while being more computationally efficient.
>
> Therefore, our analysis improves Tian et al’s in many aspects and applies to a wider context. And we achieve this with new perspectives and techniques that are quite distinctive from Tian et al’s. We have included this discussion in **Appendix F**.
>
> ---
> Thanks again for your careful reading and hope our explanations above could address your concerns. Please let us know if there is more to clarify.

---

### Official Review · Reviewer_hvBQ · 2022-11-03

**Confidence:** 4
**Clarity, Quality, Novelty And Reproducibility:** See the above comments. The evluation…
**Correctness:** 3
**Technical Novelty And Significance:** 3
**Empirical Novelty And Significance:** 2
**Recommendation:** 6

**Strength And Weaknesses:**

Pros:
1.	The paper analyzes how symmetric and asymmetric frameworks alleviate different collapses (dimensional and complete).
2.	RDM designs a new asymmetric design that also work well in practice.

Cons:
1.	Limited novelty: SymSimSiam is quite similar to the combination of DINO and SimSiam. It seems to me like just borrowing the centering and sharpening techniques of DINO to $l_2$ space (see Algorithm 1).
2.	Alignment in eigenspace is similar to whitening operation and the authors may miss some related works [1, 2]. Author may compare RDM with these methods on methodology.
3.	The limited performance on CIFAR and ImageNet. Authors claim the proposed method can get comparable results on these benchmarks. However, I don’t see any result that can lead to this conclusion (For example, with 100 epochs pretraining, RDM gets 85.7% and 58.4% accuracies on CIFAR10 and CIFAR100 datasets). Maybe the authors can comment on this.

[1] Weng X, Huang L, Zhao L, et al. An Investigation into Whitening Loss for Self-supervised Learning[J]. arXiv preprint arXiv:2210.03586, 2022.
[2] Zhang S, Zhu F, Yan J, et al. Zero-CL: Instance and Feature decorrelation for negative-free symmetric contrastive learning[C]//International Conference on Learning Representations. 2021.

**Summary Of The Paper:**

This paper proposes RDM theory, which is applicable to different asymmetric designs (with and without the predictor), and can serve as a unified understanding of existing non-contrastive learning methods. Besides, the RDM theory also provides practical guidelines for designing many new non-contrastive variants. RDM achieves comparable performance to existing methods on benchmark datasets, and some of them even outperform the baselines.

**Summary Of The Review:**

The paper has some merits while I currently still have some concerns and questions as above stated. I would like to see the authors' response.

---post rebuttal:

The authors have made good job in clarifying their novelty not only to my original concerns but also the other reviewers'. I appreciate the novelty of this work and increase my score from 5 to 6. Yet the performance advantage of this method shall be more carefully and comprehesively discussed. BYOL actually used a non-linear projector while the authors did not show their non-linear version's performance.

---

> ### Author Response · Authors · 2022-11-14
> **Response to Reviewer hvBQ (2/2)**
>
> **Q3**. The limited performance on CIFAR and ImageNet.
>
> > Authors claim the proposed method can get comparable results on these benchmarks. However, I don’t see any result that can lead to this conclusion (For example, with 100 epochs pretraining, RDM gets 85.7% and 58.4% accuracies on CIFAR10 and CIFAR100 datasets). Maybe the authors can comment on this.
>
> **A3**. We find the results that the reviewer mentioned appears in Table 4 in Appendix, and we quote it below for reference. We can see  that **our method** gets 85.7% and 58.4% on CIFAR-10 and CIFAR-100 with 100-epoch training, while **BYOL** only gets 80.81% and 49.65 **with a linear predictor**, and 82.14% and 53.92% with its original nonlinear predictor. Therefore, we can see that our method obtains a significant improvement over BYOL under 100-epoch training (5%↑ on CIFAR-10 and 9%↑ on CIFAR-100 with both linear predictors). Under longer training, e.g., 400 epochs, we can see that **our method can attain 90.73% on CIFAR-10 and 65.88% on CIFAR-100**, which is still much higher than BYOL with a linear predictor (89.73% and 60.66%), and comparable to BYOL with nonlinear predictor (91.24% and 67.41%). These results **strongly support our claim that our proposed method is comparable to BYOL**. We have revised our table caption and row names to present the results more clearer.
>
> | Method | CIFAR-10 |  | CIFAR-100 |  |
> | --- | --- | --- | --- | --- |
> |  | 100 epochs | 400 epochs | 100 epochs | 400 epochs |
> | BYOL (nonlinear predictor) | 82.14 | 91.24 | 53.92 | 67.41 |
> | BYOL (linear predictor) | 80.81 | 89.73 | 49.65 | 60.66 |
> | Ours (linear predictor, best result) | 85.73 | 90.73 | 58.40 | 65.88 |
>
> ---
>
> Thanks again for your detailed comments and encouraging score. Hope our explanations could ease your concerns. Please let us know if there is more to clarify.

---

> ### Author Response · Authors · 2022-11-14
> **Response to Reviewer hvBQ (1/2)**
>
> We thank Reviewer tWSB for your constructive comments, though there might be some misunderstandings of our contributions, as we elaborated below.
>
> ---
>
> **Q1.** Limited Novelty.
>
> > SymSimSiam is quite similar to the combination of DINO and SimSiam. It seems to me like just borrowing the centering and sharpening techniques of DINO to l2 space (see Algorithm 1).
> >
>
> **A1.** We are afraid that you might misunderstand the point that we try to make in SymSimSiam. Below we explain it in detail.
>
> - First of all, SymSimSiam is only an **illustrative example** to motivate our discussion on how asymmetric design works in Sec 4. It is **not a solution that we propose, but a phenomenon that helps us understand what we need in existing asymmetry designs.**
> - **Significance of SymSimSiam**. The major point that we try to deliver is a fresh new message to this community, that we can avoid full collapse with **symmetric vanilla architecture** (vanilla NN layers without specific modules like whitening transformations) and alignment loss alone. SymSimSiam is the **first method to attain this goal,** with provable guarantees on non-collapse (Thm 1).  Another contribution of SymSimSiam is to show that a **non-full-collapsed encoder like SymSimSiam can also suffer from dimensional collapse.** Since previous studies on non-contrastive learning only consider full collapse, e.g. Tian et al. (2021), this observation becomes the key motivation for us to study dimensional collapse in non-contrastive learning.
> - **Difference to DINO**. In contrast to SymSimSiam, DINO is an ***asymmetric*** architecture that differs from ours in the following key aspects:
>     - 1) DINO **only applies centering to the target branch**, while SymSimSiam applies to both branches;
>     - 2) DINO applies centering before (softmax) normalization, while SymSimSiam **applies centering after (l2) normalization;**
>     - 3) DINO **applies different sharpening parameters** to two branches, while SymSimSiam has no sharpening.
>
>     As a result of these key differences, **without any of the two asymmetric designs (centering and sharpening), DINO will fully collapse** (see fig7 and p17 in DINO paper). In comparison, **SymSimSiam won’t fully collapse.**
>
>
> ---
>
> **Q2**. Alignment in eigenspace is similar to whitening operation and the authors may miss some related works [1, 2]. Author may compare RDM with these methods on methodology.
>
> **A2**. Thanks for pointing out these related works. We mentioned some whitening-based methods in Related Work, and we have further included these in the revised version. These whitening-based methods generally belong to the **feature decorrelation methods**, including Barlow Twins (whitening loss), and DBN, Zero-CL (whitening transformation). Instead, in this work, the main goal of RDM is to provide **theoretical explanations and guarantees** for existing **asymmetry-based** non-contrastive methods (SimSiam, BYOL, SwAV and DINO).
>
> - From our perspective, feature decorrelation methods have a direct explanation for their non-collapsing behavior. Also, as pointed out by many people [3,4], there is still **a direct equivalence between whitening loss and uniformity loss**. Therefore, we can also say that decorrelation methods also **implicitly use negative samples**, making it easy to be understood.
> - In comparison, the non-collapsing behavior of asymmetric designs is less understood, and we cannot find clues about how negative samples would be incorporated. RDM shows that the alignment loss will have an implicit **eigenspace alignment** effect, but it alone is **not sufficient** to prevent collapse (as in SymSimSiam).
> - Instead, the key analysis of RDM is to show that the asymmetric architectures help avoid collapse by implicitly behaving as low-pass online filters (theoretically justified in Sec 5.1) and high-pass target filters (empirically justified in Sec 5.2). In particular, the target-branch transformations will implicitly behave like a generalized whitening operation. We are the first to point out this connection, and our analysis also applies to the standard whitening technique as a special case.
>
> [1] Weng X, Huang L, Zhao L, et al. An Investigation into Whitening Loss for Self-supervised Learning. NeurIPS 2022.  [https://arxiv.org/pdf/2210.03586](https://arxiv.org/pdf/2210.03586)
>
> [2] Zhang S, Zhu F, Yan J, et al. Zero-CL: Instance and Feature decorrelation for negative-free symmetric contrastive learning. ICLR. 2022. [https://openreview.net/pdf?id=RAW9tCdVxLj](https://openreview.net/pdf?id=RAW9tCdVxLj)
>
> [3]  Beyond Separability: Analyzing the Linear Transferability of Contrastive Representations to Related Subpopulations. NeurIPS 2022. (discussion on page 4) [https://arxiv.org/pdf/2204.02683.pdf](https://arxiv.org/pdf/2204.02683.pdf)
>
> [4] On the duality between contrastive and non-contrastive
> self-supervised learning. [https://arxiv.org/pdf/2206.02574.pdf](https://arxiv.org/pdf/2206.02574.pdf)

---

### Author Response · Authors · 2022-11-15
**A Summary of Paper Updates**

We thank all reviewers for your constructive comments. To better present and justify our theory, following your suggestions, **we have organized this paper (particularly Section 4),** to provide better theoretical justifications for the proposed mechanism. ***After revision, each mechanism is now supported with both theoretical justification and empirical verification***. Besides, we also made the following major changes:

- Sec 4.1: Add background on spectral filters (explain low-pass and high-pass filters).
- Sec 4.2: Add **Lemma 1 (new)** that provides theoretical justification for eigenspace alignment.
- Sec 4.2: Revise **Mechanism 1** to include the positive correlation and add empirical evidence in **Appendix E.1 (new)**.
- Sec 4.4: Revise **Theorem 4** to be clear on the assumptions and fix the bugs.
- **Appendix F (new)**: add an explicit comparison with Tian et al. (2021) on the key differences.

---

### Decision · Program_Chairs · 2023-01-20

**Decision:**

Accept: poster

**Justification For Why Not Higher Score:**

The contribution is beyond the acceptance bar but not great enough for a spotlight.

**Justification For Why Not Lower Score:**

The authors' responses have successfully addressed reviewers' concerns and we all agree to accept the paper.

**Metareview: Summary, Strengths And Weaknesses:**

All reviewers agree that this paper has met the bar for acceptance due to its technical contributions.

**Note From Pc:**

if the above contains the word "oral" or "spotlight" please see: "oral" presentation means -> notable-top-5% and "spotlight" means -> notable-top-25%. As stated in our emails, we are disassociating presentation type from AC recommendations

**Summary Of Ac-Reviewer Meeting:**

The authors' responses have successfully addressed reviewers' concerns and we all agree to accept the paper.